# Multivalency of NDC80 in the outer kinetochore is essential to track shortening microtubules and generate forces

Vladimir A Volkov[1†], Pim J Huis in 't Veld[2†], Marileen Dogterom[1]*, Andrea Musacchio[2]*

[1]Department of Bionanoscience, Faculty of Applied Sciences, Delft University of Technology, Delft, Netherlands; [2]Department of Mechanistic Cell Biology, Max Planck Institute of Molecular Physiology, Dortmund, Germany

**Abstract** Presence of multiple copies of the microtubule-binding NDC80 complex is an evolutionary conserved feature of kinetochores, points of attachment of chromosomes to spindle microtubules. This may enable multivalent attachments to microtubules, with implications that remain unexplored. Using recombinant human kinetochore components, we show that while single NDC80 complexes do not track depolymerizing microtubules, reconstituted particles containing the NDC80 receptor CENP-T bound to three or more NDC80 complexes do so effectively, as expected for a kinetochore force coupler. To study multivalency systematically, we engineered modules allowing incremental addition of NDC80 complexes. The modules' residence time on microtubules increased exponentially with the number of NDC80 complexes. Modules with two or more complexes tracked depolymerizing microtubules with increasing efficiencies, and stalled and rescued microtubule depolymerization in a force-dependent manner when conjugated to cargo. Our observations indicate that NDC80, rather than through biased diffusion, tracks depolymerizing microtubules by harnessing force generated during microtubule disassembly.
DOI: https://doi.org/10.7554/eLife.36764.001

**\*For correspondence:**
m.dogterom@tudelft.nl (MD);
andrea.musacchio@mpi-dortmund.mpg.de (AM)

[†]These authors contributed equally to this work

## Introduction

Multivalency plays a crucial role in a myriad of macromolecular interactions, often through the formation of condensates at membranes or in the cytosol (*Banani et al., 2017*; *Banjade and Rosen, 2014*; *Li et al., 2012*; *Su et al., 2016*). While it is well established that multivalent ligands may have increased binding affinity for multimeric targets in comparison to equivalent monovalent ligands, the quantitative effects of multivalency for macromolecular interactions cannot be easily predicted (*Jencks, 1981*), and needs therefore to be determined experimentally.

Kinetochores are multiprotein assemblies built on the centromeres of chromosomes (*Musacchio and Desai, 2017*). Their ability to hold on to the ends of microtubules, dynamic polymers that alternate between phases of growth and shrinkage, is crucial for the segregation of replicated chromosomes (sister chromatids) to the daughter cells during cell division. This event does not require molecular motors, and chromosome propulsion is mediated by the ability of kinetochores to remain attached to depolymerizing microtubules (*Grishchuk and McIntosh, 2006*; *Tanaka et al., 2007*). With hundreds of different proteins assembled on a chromosome's centromeric region, kinetochores are the epitome of a multivalent proteinaceous platform, but how the stoichiometry and the modular organization of kinetochore components contribute to microtubule end-coupling remains poorly understood.

**eLife digest** Before a cell divides, its genome duplicates so that each copy can be given to the daughter cells. In a dividing cell, the chromosomes – the structures that store genetic information – look like an 'X'. This is because each chromosome is formed of two identical, rod-like, 'sister chromatids' which are attached by their middle. Each daughter cell should inherit one of the chromatids.

As division progresses, both sister chromatids in a pair fasten to 'microtubules', string-like structures made of a large number of identical proteins stacked together. These strings attach each chromatids to opposite sides of the cell. Then, the ends of the microtubules that bind to a chromatid start to peel off and disassemble. The microtubules get shorter and shorter, which creates a force that pulls the chromatids apart.

Microtubules latch on a chromatid via a large structure known as the kinetochore, which has tether-like protein complexes called NDC80 at its surface. NDC80 links the kinetochore with the microtubules, yet little is known about this connection. In particular, it is unclear how this complex relays the forces from the shortening microtubules to the chromatids, and how many NDC80 complexes are required for this process.

To study how these proteins interact without any molecular background 'noise' from the cell, Volkov, Huis in 't Veld et al. engineered simplified versions of the microtubule-kinetochore-NDC80 connection using components of human kinetochores. These versions, named 'modules', contained different numbers of NDC80 complexes, from one to four copies.

Volkov, Huis in 't Veld et al. found that single NDC80 complexes did not follow the microtubules as they shortened, while the connections with two or more NDC80 complexes did. When a few modules, each with two or three NDC80s, were closeby, they also bound to the end of the same shortening microtubule, and captured more force as a team. NDC80 complexes therefore work together to connect to microtubule ends and harness their energy.

The artificial kinetochore-microtubule-NDC80 connections developed by Volkov, Huis in 't Veld et al. provides a new method to study how cells divide, and it could reveal how other proteins and biological processes participate in this mechanism. It could also help understand how chromatids are kept from separating incorrectly during division, which is an error that could be fatal for the cell.
DOI: https://doi.org/10.7554/eLife.36764.002

The 4-subunit NDC80 complex (NDC80, for nuclear division cycle 80 complex) provides the crucial link between kinetochores and microtubules. In vitro reconstitution reveals that a single NDC80 complex is unable to follow dynamic microtubules (*Powers et al., 2009*; *Schmidt et al., 2012*). Microtubule end-tracking activity is conferred to NDC80 by the presence of Dam1 or Ska complexes, which provide additional binding sites to microtubule walls and dynamic ends (*Tien et al., 2010*; *Lampert et al., 2010*; *Schmidt et al., 2012*). NDC80 was also shown to gain tip-tracking properties in the presence of factors that trigger multimerization, such as the functionalized surface of beads (*McIntosh et al., 2008*) or the presence of an antibody (*Powers et al., 2009*). These observations were interpreted as being supporting a model for NDC80 interaction with depolymerizing microtubules known as biased-diffusion (*Powers et al., 2009*). Biased diffusion models assume that a connected ensemble of diffusive, low-affinity microtubule binding elements undergoes directional Brownian motion on a shortening microtubule if it can maximize its binding energy by retaining as many as possible of the binding elements bound to the microtubule lattice. If the lattice is shrinking, this will in turn act against detachment and will promote depolymerization-connected motion (*Asbury et al., 2011*; *Grishchuk, 2017*).

An alternative to biased diffusion models is a category of models in which kinetochores harness the energy released by a depolymerizing microtubule end to move together with it. In opposition to the biased-diffusion models, these models predict that the microtubule binding elements of the kinetochore are poorly diffusive and high-affinity, and that a considerable part of the energy released during microtubule depolymerization must be used to translate the kinetochore binders along the lattice (*Asbury et al., 2011*; *Grishchuk, 2017*). This, in turn, is predicted to slow down microtubule depolymerization, a feature that biased-diffusion models do not predict. Until now,

none of the predictions that would allow distinguishing which of these model classes best applies to the NDC80 complex has been rigorously tested.

How NDC80 is recruited to the kinetochore is well understood. First, NDC80 binds the 4-subunit MIS12 complex, whose kinetochore recruitment, in turn, depends on the chromatin proximal kinetochore subunits CENP-C and CENP-T (*Huis In 't Veld et al., 2016*; *Petrovic et al., 2016*; *Przewloka et al., 2011*; *Screpanti et al., 2011*). Second, NDC80 is recruited by direct binding to two phosphorylated motifs near the N-terminus of CENP-T (*Huis In 't Veld et al., 2016*; *Gascoigne et al., 2011*; *Kim and Yu, 2015*; *Suzuki et al., 2015*; *Malvezzi et al., 2013*; *Nishino et al., 2012*). In agreement with these multiple binding modes, quantitative fluorescence microscopy and bottom-up reconstitutions indicated that kinetochores contain multiple NDC80 complexes, with recent estimates converging on eight complexes per microtubule-binding site (*Huis In 't Veld et al., 2016*; *Suzuki et al., 2015*; *Weir et al., 2016*). In humans, a single kinetochore contains a molecular lawn of approximately 200 NDC80 complexes that bind 15–20 microtubules (*Suzuki et al., 2015*; *Wendell et al., 1993*). In principle, this distribution appears ideally suited to exploit the potential of multivalency, because multiple NDC80 complexes may bind concomitantly to the same microtubule. However, how the stoichiometry and modular organization of NDC80 contribute to the coupling of kinetochores to microtubules remains unclear. To address the role of multivalency at the kinetochore-microtubule interface quantitatively, we reconstituted kinetochore modules with multiple NDC80 complexes bound to phosphorylated CENP-T or to a streptavidin-based multimerization system. The multivalency of NDC80 modules is essential for these modules to interact with dynamic microtubules. We report the forces generated when shortening microtubules pull on these assemblies and illustrate how tension might modulate microtubules dynamics. Collectively, our results strongly suggest that NDC80 harnesses the energy of depolymerizing microtubules.

## Results

### Multivalent NDC80 binding to CENP-T:MIS12 promotes motility with microtubule ends

We have previously shown that CENP-T that is phosphorylated by CDK1:Cyclin-B can recruit NDC80 complexes to p-Thr11 and p-Thr85 and a MIS12:NDC80 complex to p-Ser201 (*Huis In 't Veld et al., 2016*). To test how phosphorylated CENP-T$^{2-373}$, hereafter called CENP-T, that is associated with multiple NDC80 complexes interacts with dynamic microtubules, protein complexes labeled with single fluorophores were imaged on dynamic microtubules using TIRF microscopy (*Figure 1A*). Consistent with previous findings (*Powers et al., 2009*; *Schmidt et al., 2012*), monomeric NDC80 complexes briefly interacted with the microtubule lattice but did not follow the ends of depolymerizing microtubules (*Figure 1B*). In contrast, tip-tracking events were observed for CENP-T$^{Alexa488}$:NDC80 and CENP-T$^{Alexa488}$:MIS12:NDC80 complexes that were purified by size-exclusion chromatography and then added to dynamic microtubules (*Figure 1—figure supplement 1*, *Figure 1—figure supplement 2A*). As predicted, this binding strictly depended on the presence of NDC80; a similarly prepared stoichiometric complex of MIS12 and phosphorylated CENP-T$^{Alexa488}$ did not bind microtubules over a 35-fold range of concentrations (*Figure 1C* and *Figure 1—figure supplement 2B*). When we combined NDC80$^{TMR}$ and CENP-T$^{Alexa488}$:MIS12, co-localization of fluorescence that prevailed at the end of shortening microtubules was observed (*Figure 1D* and *Figure 1—figure supplement 3A*). In a significant number of cases, we observed tip-tracking NDC80$^{TMR}$ that apparently lacked CENP-T$^{Alexa488}$:MIS12. Since monomeric NDC80$^{TMR}$ under similar conditions did not result in tip-tracking (*Figure 1B*), we suspect that this tip-tracking subset contains bleached CENP-T$^{Alexa488}$:MIS12. In order to assess the composition of CENP-T$^{Alexa488}$:MIS12:NDC80$^{TMR}$, we normalized the intensities of Alexa488 and TMR signals to the intensity of a single fluorophore, as determined from photobleaching events (*Figure 1E*). Tip-tracking assemblies contained 1.5 ± 0.7 CENP-T molecules (mean and SD), 5 ± 2 NDC80 molecules (n = 27), and most complexes contained three NDC80 complexes per CENP-T:MIS12, a number that is in agreement with the biochemically determined number of NDC80 binding sites on phosphorylated CENP-T:MIS12. (*Figure 1F*). These assemblies were formed when CENP-T:MIS12 was added in a 100-fold molar excess over NDC80. When we added both complexes at the equimolar N/TM ratio, assemblies with a larger number of NDC80 complexes

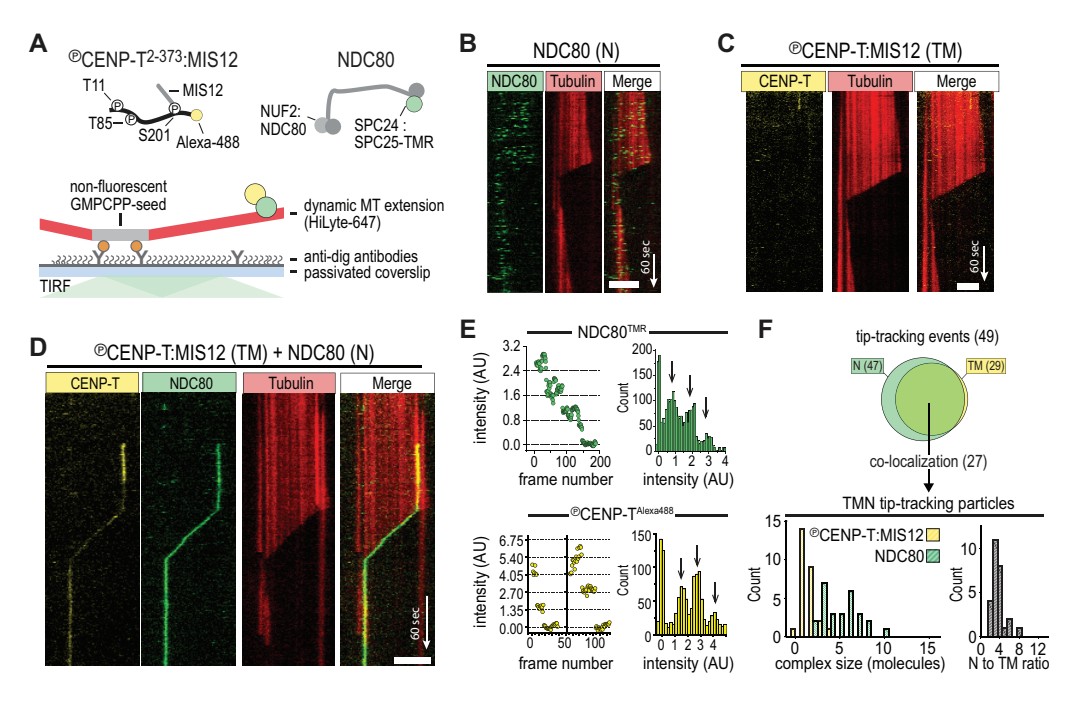

**Figure 1.** CENP-T-mediated oligomerization of NDC80 produces particles that follow microtubule disassembly. (a) Schematic representation of Alexa-488-labeled, phosphorylated CENP-T:MIS12 and TMR-labeled NDC80 in a single-molecule TIRF setup with dynamic microtubules. (b) Monomeric NDC80 at 200 pM does not follow microtubule shortening. (c) MIS12-CENP-T at 2 nM does not bind to microtubules and does not follow microtubule shortening. (d) MIS12:CENP-T$^{Alexa488}$ (2 nM, yellow) and NDC80$^{TMR}$ (20 pM, green) mixed in the flow-chamber, co-localize in a particle that follows the shortening end of a microtubule. (e) Left: time traces showing photobleaching of NDC80$^{TMR}$ (green) and CENP-T$^{Alexa488}$ particles (yellow). Note that the CENP-T particle initially bleaches two fluorophores simultaneously. Right: distributions containing all bleaching traces (n = 13 for TMR and n = 20 for Alexa488). Arrows show intensities of 1, 2 and 3 fluorophores. See Materials and methods for details. (f) Euler diagram of tip-tracking particles produced by mixing NDC80 and MIS12:CENP-T (top). Distributions of initial, unbleached fluorescence intensities (expressed as the number of fluorophores) of NDC80$^{TMR}$ (green) and MIS12:CENP-T$^{Alexa488}$ (yellow) in the tip-tracking particles containing both signals (bottom left). Black bars show ratio of NDC80 to CENP-T (bottom right). Scale bars: 5 μm (horizontal), 60 s (vertical).

DOI: https://doi.org/10.7554/eLife.36764.003

The following figure supplements are available for figure 1:

**Figure supplement 1.** Reconstitution of NDC80 and MIS12 on CENP-TAlexa-488 phosphorylated by CDK1:Cyclin-B.

DOI: https://doi.org/10.7554/eLife.36764.004

**Figure supplement 2.** NDC80 promotes tip-tracking of reconstituted TN and TMN.

DOI: https://doi.org/10.7554/eLife.36764.005

**Figure supplement 3.** Tip-tracking by NDC80 oligomerized on CENP-T:MIS12.

DOI: https://doi.org/10.7554/eLife.36764.006

and an increased tendency to aggregate were observed (*Figure 1—figure supplement 3B–C*). Collectively, these results indicate that the combination of multiple NDC80 complexes on the CENP-T:MIS12 platform and the abundance of binding sites for the NDC80 complex on the microtubule lattice promote the formation of multivalent complexes. A confounding factor emerging from our analysis, however, is that the number of NDC80 complexes in the CENP-T:MIS12 assemblies appeared to vary significantly between complexes (*Figure 1F* and *Figure 1—figure supplement 3*), complicating our attempts to perform a rigorous analysis of the role of multivalency in kinetochore-microtubule attachment. While the reconstituted CENP-T$^{Alexa488}$:MIS12:NDC80$^{TMR}$ complexes have very well defined stoichiometry at the typical low micromolar concentrations of our biochemical analyses (*Huis In 't Veld et al., 2016*), variability of stoichiometry at the picomolar concentrations required to observe these complexes on microtubules at single-molecule level may reflect the absence of additional (and currently unknown) stabilizing interactions or post-translational modifications.

## Incremental addition of NDC80 results in hyperstable microtubule binding

Therefore, in order to control stoichiometry more precisely, we set out to engineer a scaffold that can covalently bind a defined number of NDC80 complexes. Traptavidin (T), a streptavidin variant with a biotin dissociation constant in the low femtomolar range (*Chivers et al., 2010*), and a biotin-binding deficient streptavidin variant tagged with a SpyCatcher module (S). T and S were folded into tetramers with stoichiometries varying from $T_4S_0$ to $T_0S_4$ and separated from each other by ion-exchange chromatography (*Fairhead et al., 2014*; *Zakeri et al., 2012*) (*Figure 2—figure supplements 1* and *2*). Purified TS tetramers were covalently coupled to NDC80 complexes with SPC25$^{TMR}$ and SPC24$^{SpyTag}$ (*Figure 2A*). Assemblies containing one, two, three, or four NDC80 complexes were subsequently separated by size-exclusion chromatography and analyzed by SDS-PAGE. Note that the TS tetramers only denature in SDS-containing sample buffer after extensive boiling (*Figure 2—figure supplement 1*). Inspection of the purified TS-NDC80 modules by electron microscopy after glycerol spraying and low-angle metal shadowing confirmed the integration of the highly elongated (~60 nm) NDC80 tethers (*Ciferri et al., 2008*; *Huis In 't Veld et al., 2016*; *Wei et al., 2005*) on a central TS density (*Figure 2B–C*). The overall appearance of these artificial particles, including the flexible orientation of the NDC80 complexes, is highly reminiscent of the multiple NDC80 complexes bound to CENP-T:MIS12 (*Figure 2D*) (*Huis In 't Veld et al., 2016*). We thus reconstituted a proxy of the outer kinetochore with precise control over the number of incorporated NDC80 complexes.

To test at a single-molecule level how assemblies with a precisely defined NDC80 stoichiometry interact with microtubules, we measured the residence time of TS-NDC80 modules on taxol-stabilized microtubules using total internal reflection fluorescence (TIRF) microscopy (*Figure 2E* and *Figure 2—figure supplement 3*). The residence time of these assemblies on microtubules increased more than ten-fold for every additional NDC80 complex (*Figure 2E*). The residence time of CENP-T: NDC80 assemblies (*Figure 1—figure supplement 1*) under similar conditions ranged from multiple seconds to multiple minutes (*Figure 2—figure supplement 3*). This is comparable with divalent and trivalent NDC80 TS-modules and supports the idea that NDC80 that is bound to TS modules and NDC80 that is bound to CENP-T interact with microtubules in a similar manner.

To simulate the binding of TS-NDC80 assemblies to microtubules in silico, we assumed transitions between the number of microtubule-bound NDC80 tethers based on the $k_{on}$ and $k_{off}$ rates of each individual NDC80. The initial landing rate of an assembly on a microtubule was ignored so that each simulation started with a single NDC80 bound to a microtubule and stopped after detachment of all NDC80s. Using the reciprocal of the residence time of $T_3S_1[NDC80]_1$ as $k_{off}$, stochastic simulations of the residence times of mono-, di-, tri-, and tetravalent assemblies were used to determine $k_{on}$ (*Figure 2E*). A fit to the data resulted in a $k_{on}$ of 2.6 s$^{-1}$. Assuming a TS-NDC80 assembly as a sphere, one NDC80 inside the sphere has a local concentration of 1.7 µM. This predicts a concentration-dependent association rate of 1.5 µM$^{-1}$s$^{-1}$, a value that falls well within the published range of $k_{on}$ for free NDC80 binding to microtubules (*Powers et al., 2009*; *Tien et al., 2010*; *Zaytsev et al., 2015*). Taken together, this demonstrates that all NDC80 complexes in TS-NDC80 modules can interact with one microtubule and illustrates how NDC80 multivalency stabilizes microtubule binding.

## NDC80 multivalency promotes cargo motility at shortening microtubules

We next used dynamic microtubules to compare CENP-T- and TS-based oligomerization strategies for the assembly of tip-tracking oligomers of NDC80. Di-, tri-, as well as tetravalent TS-NDC80 assemblies tracked shortening microtubule tips and each additional NDC80 increased the fraction of tip-tracking assemblies (*Figure 3A–C* and *Figure 3—figure supplement 1A–D*). Free microtubules in the same flow chamber shortened faster than the ones with TS-NDC80 at the tip (*Figure 3D*). CENP-T- and CENP-T:MIS12-mediated oligomers of NDC80 slowed down microtubule shortening in a similar way (*Figure 3—figure supplement 1E–F*). The ability of a single particle to slow down microtubule depolymerization suggests that tip-tracking NDC80 oligomers form a direct connection to the shortening microtubule ends.

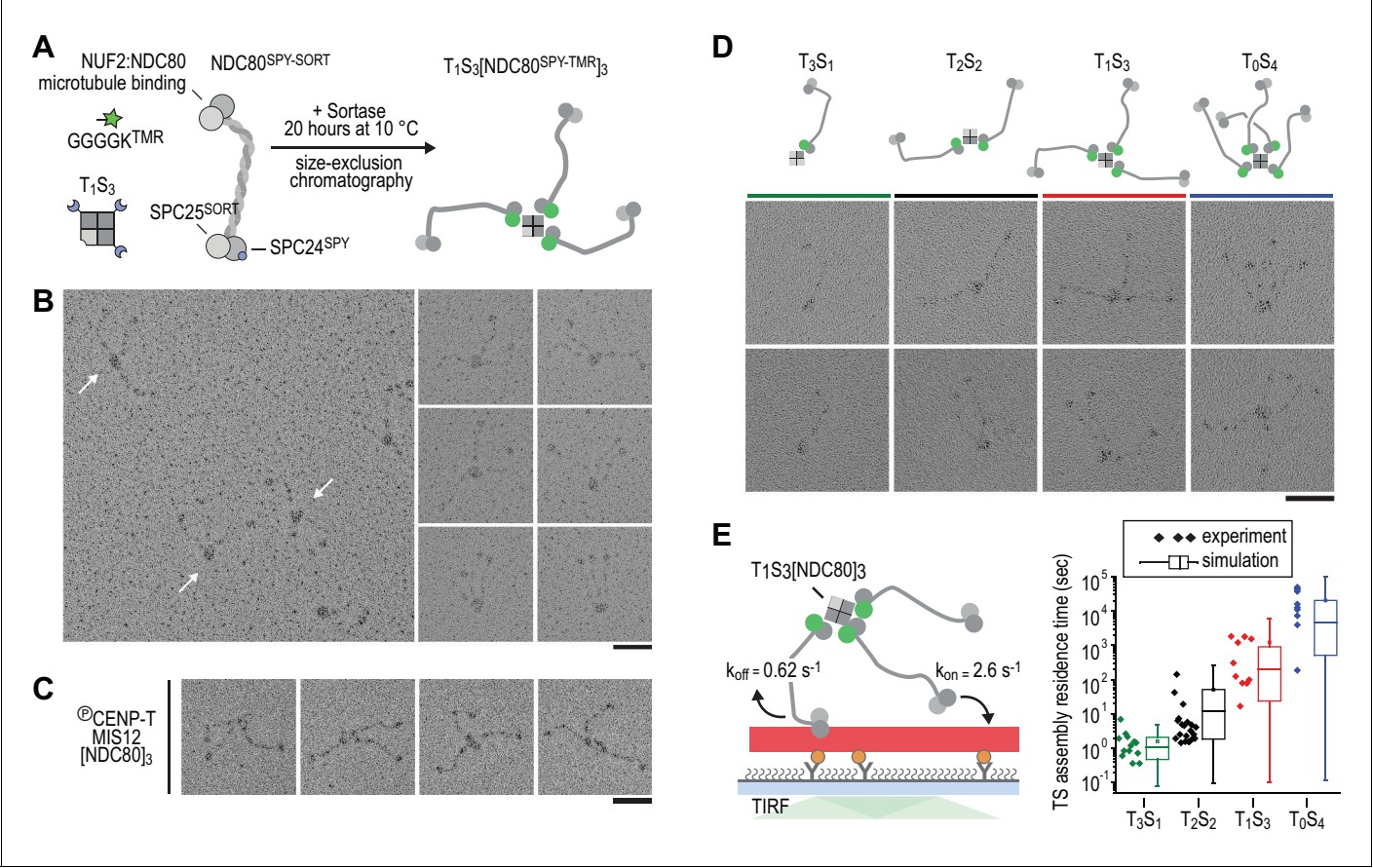

**Figure 2.** Incremental addition of NDC80 results in hyperstable microtubule binding. (a) NDC80$^{SPY-SORT}$ was fluorescently labelled and covalently bound to TS assemblies. The cartoon shows the formation of T$_1$S$_3$[NDC80]$_3$ assemblies. Size-exclusion chromatography and SDS-PAGE analysis are shown in *Figure 1—figure supplement 1*. (b) T$_1$S$_3$[NDC80]$_3$ assemblies were analysed by electron microscopy after low-angle rotary shadowing. Three flexible NDC80 complexes of approximately 60 nm originate from central T$_1$S$_3$ densities (white arrows in the field of view) Scale bar 50 nm. (c) Representative micrographs of CENP-T:MIS12:[NDC80]$_3$ as analysed previously (*Huis In 't Veld et al., 2016*). (d) Side-by-side comparison of NDC80 coupled to T$_3$S$_1$, T$_2$S$_2$, T$_1$S$_3$, and T$_0$S$_4$. Cartoons represent the approximate orientation of assemblies in the upper row of micrographs. Scale bar 50 nm. (e) Residence time of quantized NDC80 assemblies on taxol-stabilized microtubules as determined experimentally (dots) and as predicted by a series of 1000 simulations (box and whiskers plot; box: 25–75%, horizontal line: median, whiskers: 5–95%). NDC80 complexes of a microtubule-bound TS-NDC80 assembly attach to and detach from microtubules with rates of $k_{on}$ and $k_{off}$, respectively. The residence time of an oligomer is defined as the time between the association of its first NDC80 tether and the detachment of all NDC80 tethers.

DOI: https://doi.org/10.7554/eLife.36764.007

The following figure supplements are available for figure 2:

**Figure supplement 1.** A reconstituted system to precisely control NDC80 stoichiometry (Part I).
DOI: https://doi.org/10.7554/eLife.36764.008

**Figure supplement 2.** A reconstituted system to precisely control NDC80 stoichiometry (Part II).
DOI: https://doi.org/10.7554/eLife.36764.009

**Figure supplement 3.** Characterization of oligomerized NDC80 on taxol-stabilized microtubules.
DOI: https://doi.org/10.7554/eLife.36764.010

To test if cargo-bound assemblies retain their ability to bind microtubules, we prepared biotinylated nanogold particles conjugated to trivalent TS-NDC80. These particles localized on or in between microtubules, indicating that biotinylated cargo was coupled to microtubules via TS-NDC80 modules (*Figure 3E* and *Figure 3—figure supplement 2*). To test if TS-NDC80 can couple forces generated by the protofilament depolymerization to the movement of cargo, we attached a biotinylated glass bead amenable to optical trapping to the biotin-binding traptavidin (T) of TS-NDC80 modules (*Figure 3E*). Binding of TS-NDC80 resulted in beads that were able to track shortening (*Figure 3F*) or -in rare cases- growing (*Figure 3G*) microtubule ends.

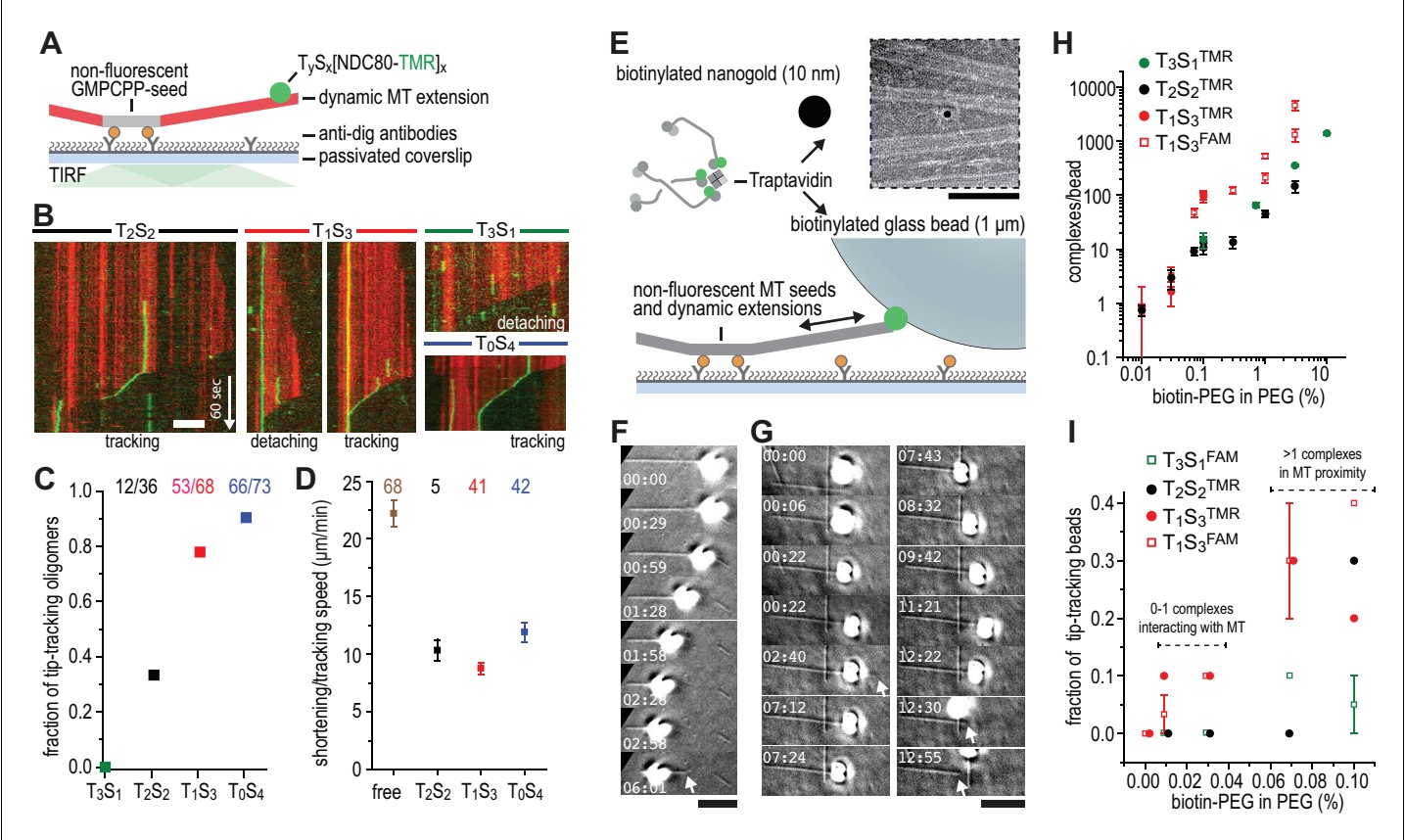

**Figure 3.** Trivalent TS-NDC80 efficiently tracks depolymerizing microtubules and transports cargo. (a) Schematic representation of the experimental setup. (b) Kymographs showing NDC80 (green) assembled on $T_2S_2$, $T_1S_3$, or $T_0S_4$ tracking a depolymerizing microtubule (red). An example of a $T_1S_3[NDC80]_3$ complex that detached from the tip of shortening microtubule is also included. Scale bar 5 μm. See *Figure 3—figure supplement 1* and *Video 1*. (c) The fraction of NDC80 assemblies that track depolymerizing microtubules. (d) Comparison between microtubule depolymerization in the presence and absence of TS-NDC80 following the shortening tips. Data are shown as mean ± SEM. e) Biotinylated glass beads or nanogold particles can be conjugated to traptavidin in TS-NDC80C assemblies. Nanogold particles coated with $T_1S_3[NDC80]_3$ bound to microtubules as observed by negative-staining EM (see also *Figure 3—figure supplement 2*). Scale bar 100 nm. (f–g) Examples of glass beads coated with $T_1S_3[NDC80]_3$ tracking depolymerizing microtubules. The bead in panel g follows the growing microtubule after a rescue event until it detaches during a second depolymerization phase. White arrows indicate the dynamic microtubule tips. Scale bar 5 μm. (h) Fluorescence-based quantification of the number of complexes on glass beads coated with increasing amounts of PLL-PEG-biotin. (i) The fraction of beads coated with various TS-NDC80 assemblies that track depolymerizing microtubules as a function of the amount of biotin-PEG added to the beads.

DOI: https://doi.org/10.7554/eLife.36764.011

The following source data and figure supplements are available for figure 3:

**Source data 1.** Tip-tracking events for differently coated beads.
DOI: https://doi.org/10.7554/eLife.36764.014
**Figure supplement 1.** Characterization of oligomerized NDC80 on dynamic microtubules.
DOI: https://doi.org/10.7554/eLife.36764.012
**Figure supplement 2.** Negative-stain EM of microtubules and nanogold particles coated with $T_1S_3[NDC80]_3$.
DOI: https://doi.org/10.7554/eLife.36764.013

To control the number of TS-NDC80 assemblies on the beads, we coated beads with a mixture of poly-L-lysine-polyethylene glycol (PLL-PEG) and PLL-PEG-biotin. Using the fluorescence of NDC80C$^{TMR}$ or NDC80C$^{FAM}$ as a readout, we observed that the number of complexes on a bead's surface increased linearly from <2 complexes at 0.01% PEG-biotin to several thousand complexes at 10% PEG-biotin (*Figure 3H*). With a diameter of 1 μm, coating of a glass bead with 0.01–0.03% biotin-PEG predicts a single TS-NDC80 module interacting with a microtubule: even at the higher estimate of 5 TS modules per bead at 0.03% biotin-PEG (*Figure 3H*), the linear distance between complexes is approximately 775 nm. At biotin-PEG concentrations above 0.07% (average densities

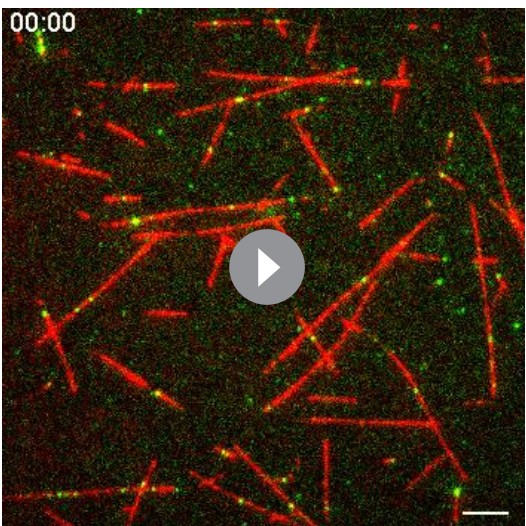

**Video 1.** Single-molecule TIRF microscopy of TS-NDC80 modules on dynamic microtubules. 35 pM of $T_0S_4[NDC80^{TMR}]_4$ (green) in the presence of 8 μM tubulin labelled with HiLyte-642 (red) and in the absence of KCl. The two-channel images were acquired every 1.1 s (shown at 30 fps). Top left corner shows time in min:sec. White arrows mark tip-tracking events. Scale bar 5 μm.

DOI: https://doi.org/10.7554/eLife.36764.015

of 10–50 TS modules per bead – see **Figure 3H**), the linear distance between adjacent complexes is 500–250 nm and multiple modules could reside in the proximity of one microtubule (see also **Powers et al., 2009**) for calculations). The presence of multiple NDC80 tethers in the proximity of a single microtubule resulted in beads that followed dynamic microtubule tips for trivalent $T_1S_3[NDC80]_3$ (13/45 cases), but occasionally also for monovalent $T_3S_1[NDC80]_1$ (2/30 cases) (**Figure 3I** and **Figure 3—source data 1**). This is consistent with the reported tip-tracking of beads that are densely decorated with monomeric NDC80 (**McIntosh et al., 2008**; **Powers et al., 2009**). Beads with the lowest biotin densities coated with monovalent $T_3S_1[NDC80]_1$ or divalent $T_2S_2[NDC80]_2$ were unable to follow microtubule tips (0/50 cases). The same coating densities for $T_1S_3[NDC80]_3$ resulted in 5/70 beads following dynamic microtubule tips (**Figure 3I** and **Figure 3—source data 1**). This suggests that a single trivalent $T_1S_3[NDC80]_3$ assembly is sufficient to couple microtubule shortening to cargo transport.

## Multivalent NDC80 stalls and rescues microtubule shortening under force

We next set out to probe the force-coupling properties of TS-NDC80 assemblies with an optical trap. For this purpose, a bead coated with TS-NDC80 was positioned near the end of a dynamic microtubule, confirmed to bind the microtubule with the trap switched off, and monitored with the trap switched on while the tip of the shortening microtubule reached the bead (**Figure 4A**). Displacement of the bead from the centre of the trap was recorded with a quadrant photo detector (QPD). A peak in the QPD signal along the microtubule's direction indicates that the force generated by the depolymerizing microtubule is converted into the displacement of the bead coated with TS-NDC80 (**Figure 4B** and **Video 2**). After initial movement with the shortening microtubule tips, the beads stalled for an average of 1.5 ± 0.2 s (mean and SEM, n = 91). We interpreted these stalls as the time in which the microtubule-generated force is counterbalanced by the force acting to return the bead to the centre of the trap, thus reducing the depolymerization speed to zero. Beads at the ends of stalled microtubules either detached from the microtubule and snapped back into the centre of the trap, or rescued microtubule depolymerization and gradually moved back to the centre of the trap along with the growing microtubule (**Figure 4C** and **Video 3**). In 13 out of 104 traces the bead detached from the microtubule before stalling; these traces were not analysed further.

We next tested the relation between the number of bead-bound TS-NDC80 modules and the stalling forces. Single trivalent $T_1S_3[NDC80]_3$ stalled microtubule depolymerization at about 1.5 piconewton (pN). Stalling forces increased to maximal values of 5–6 pN with 100–1000 $T_1S_3[NDC80]_3$ modules per bead, but did not increase further with thousands of such assemblies on a single bead (**Figure 4D**). A bead with 100–1000 TS modules on the surface has an estimated minimum of 4 $T_1S_3[NDC80]_3$ modules in the proximity of a microtubule end. This organization might be optimal to couple the energy from microtubule depolymerization into cargo movement: increasing the local NDC80C concentration on the beads beyond that did not result in higher stalling forces.

Pooling of stall forces from beads coated with varying surface densities of $T_1S_3[NDC80]_3$ and $T_2S_2[NDC80]_2$ resulted in 51 detachment events (71%) and 21 rescue events (29%) and revealed a striking correlation between the amplitude of the stalling force and the probability of a rescue event (**Figure 4E**). Interestingly, even a very dense coating of monovalent $T_3S_1[NDC80]_1$ on the beads did

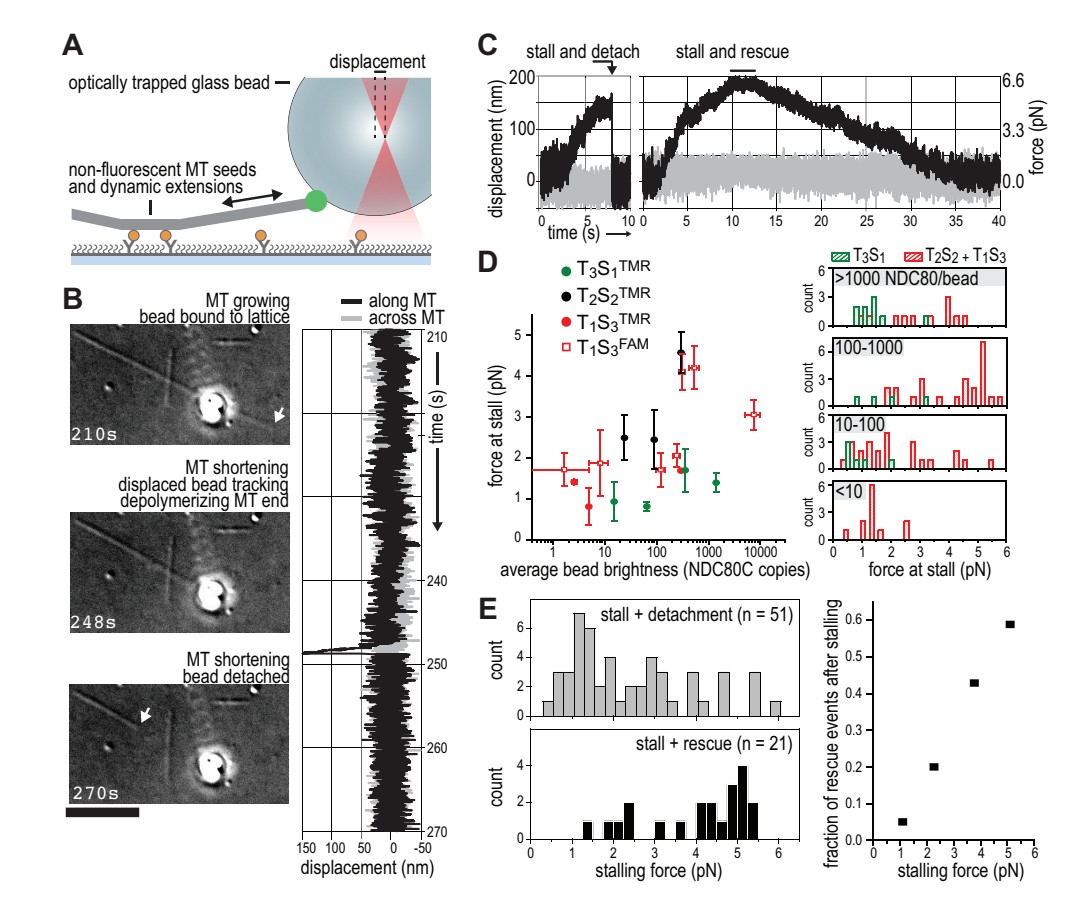

**Figure 4.** TS-NDC80 modules stall and rescue microtubule depolymerization. (a) The displacement of an optically trapped glass bead can be used to determine the force exerted by a shortening microtubule on a bead-bound TS-NDC80 oligomer. (b) Example of a trapped glass bead that is displaced along the microtubule axis as it holds on to a depolymerizing microtubule (248 s). Arrows point to the dynamic microtubule tip before (210 s) and after (270 s) the force development. The graph on the right shows unfiltered QPD signal along and across the microtubule axis. (c) Examples of unfiltered QPD signals recorded during microtubule shortening. Stalling of microtubule depolymerization by the coupled bead in the optical trap is followed by detachment of the bead (left) or a rescue of microtubule growth (right). (d) Average forces at which differently coated beads stall shortening microtubules. Data are shown as mean ±SEM. (e) Distribution of stalling forces that were followed by bead detachment from the microtubule (grey bars) or microtubule rescue (black bars). These distributions were used to calculate the fraction of events leading to a force-induced rescue (right).
DOI: https://doi.org/10.7554/eLife.36764.016

not generate stalling forces higher than 3 pN (*Figure 4D*) and never produced rescue events. Taken together, these results show that the multivalent NDC80 modules promote efficient coupling of microtubule-generated force as well as force-dependent rescue of microtubule shortening.

## The linkage between NDC80 oligomers and the microtubule end stiffens during depolymerization stalls

The ability of TS-NDC80 modules to slow down, stall, and rescue microtubule depolymerization suggests that they interact with the very end of the shortening microtubule. To further assess the mechanical properties of this interaction, we analysed the thermal fluctuations of trapped beads before and during contact with the tip of the microtubule. For a trapped bead in solution, these fluctuations are limited by the stiffness of the trap. For a trapped bead attached to a microtubule, these fluctuations additionally reflect the mechanical properties of the microtubule-bead linkage. During stalling, fluctuations along the microtubule were dampened compared to the signal across the microtubule and compared to fluctuations before and after the microtubule pulled on the bead (*Figure 5A*). This demonstrates an effective stiffening of the link between the bead and the microtubule under force. To test if pulling on microtubules and NDC80 also altered fluctuations of beads

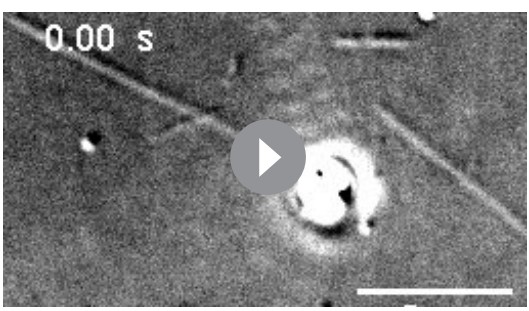

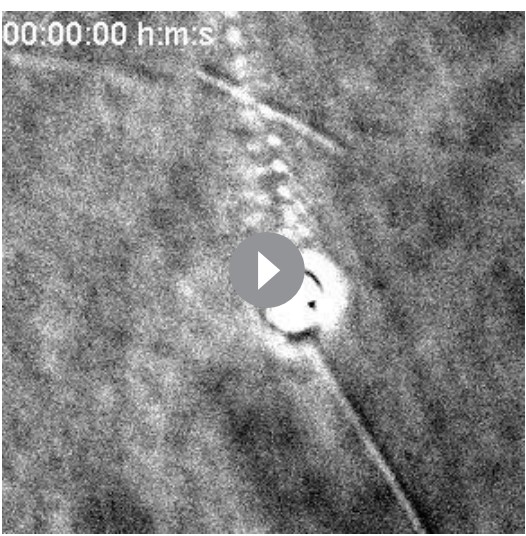

**Video 2.** A disassembling microtubule tip pulls on a trapped bead. A bead coated with 3% PLL-PEG-biotin and then saturated with $T_2S_2[NDC80^{TMR}]_2$ was attached to a microtubule with a trap (see also **Figure 3B** for still images and a complete QPD trace of this signal). Images were acquired at eight fps in DIC, background subtracted, and each 10 consecutive frames averaged (see Materials and methods). The timer at the bottom left corner shows elapsed time in seconds. The microtubule continues growing until about 230 s, when it switches to shortening and pulls on the bead at 248 s (evident from the bead's brief displacement in the direction of microtubule disassembly). From 265 s till 290 s the microtubule end is seen disassembling to the left of the bead. Scale bar 5 μm.
DOI: https://doi.org/10.7554/eLife.36764.017

**Video 3.** A microtubule is rescued five times at the bead attachment site. A bead coated with 0.3% PLL-PEG-biotin and then saturated with $T_1S_3[NDC80^{FAM}]_3$ was attached to a microtubule with a trap. The microtubule experiences dynamic instability, but its shortening is five times in a row rescued at the attached bead (at 2, 9, 24, 47 and 60 min from the start of the experiment, see timer in the top left corner). Note that each rescue is preceded by a displacement of the bead along the microtubule axis in the direction of disassembly. After 63 min, the trap stiffness is increased from initial 0.03 pN/nm to 0.13 pN/nm to manually remove the bead from the microtubule. The microtubule without the attached bead depolymerizes without stalling or rescue events. Scale bar 5 μm.
DOI: https://doi.org/10.7554/eLife.36764.018

that do not interact with the depolymerizing ends, we attached beads decorated with $T_1S_3[NDC80]_3$ assemblies either to the lattice of a microtubule away from the dynamic end, or to the stabilized end of a microtubule with a GMP-CPP cap (**Figure 5A**). Force in these experiments was exerted by pulling on the coverslip-attached microtubule seed using the piezo stage in the same direction as a depolymerizing microtubule would pull. Stiffening of the bead-microtubule link under force was observed in all cases, but was on average two times higher during stalls produced by TS-NDC80 modules interacting with depolymerizing microtubules (**Figure 5B**). Apparently, force-induced stiffening of the link provided by TS-NDC80 modules is enhanced when NDC80 molecules are interacting with depolymerizing, presumably flared protofilaments, compared to when they are interacting with straight protofilaments.

## Discussion

Productive coupling of microtubule-generated force to kinetochore motion is achieved by balancing the strength of the coupler's binding to the microtubule and the amount of energy the microtubule spends to move the coupler (**Efremov et al., 2007**). We show here that at the typical picomolar concentrations of single-molecule analyses, monomeric NDC80 complexes bind weakly to the microtubules and have no detectable effect on microtubule dynamics. Instead, the integration of two NDC80 molecules in a single module is sufficient to track the tip of a shortening microtubule and to slow down microtubule depolymerization (**Figure 3D**). A similar effect on microtubule disassembly had been previously observed using very high concentrations of monomeric NDC80 that resulted in complete decoration of the microtubule lattice with NDC80 (**Umbreit et al., 2012**).

Models of kinetochore-microtubule attachment during microtubule disassembly can be classified under two fundamental categories: (1) attachments that do not directly harness the energy liberated by a depolymerizing microtubule to generate movement, and (2) attachments that do. The 'biased

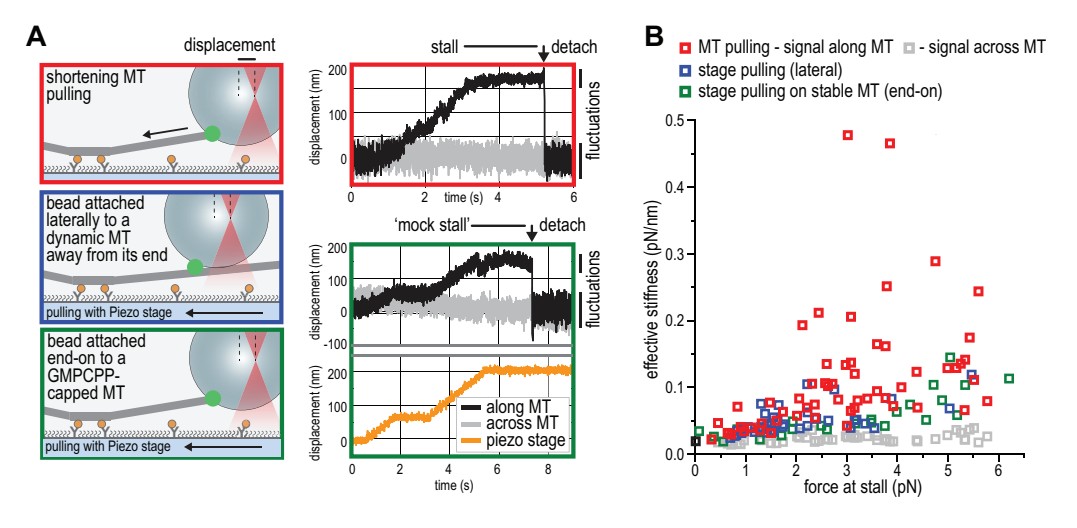

**Figure 5.** NDC80C oligomers stall microtubules through interaction with the shortening microtubule end. (a) Experimental setup to compare forces generated by shortening microtubules (red box) with forces generating by a moving stage while a bead with $T_1S_3[NDC80]_3$ is attached laterally to a dynamic microtubule (blue box), or end-on to a stabilized microtubule (green box). Examples of unfiltered QPD signals recorded during force generation are shown on the right. (b) Effective stiffness of the link between the bead and the microtubule increases with force.
DOI: https://doi.org/10.7554/eLife.36764.019

diffusion' mechanism of attachment to a depolymerizing microtubule belongs to the first category. By assuming that weakly bound couplers diffuse to explore the surface of the microtubule by steps that are driven by thermal energy, this model predicts that net kinetochore movement is not linked to an energy expenditure by the depolymerizing microtubule. Consequently, this type of attachment does not stall microtubule disassembly and is expected to have negligible effects on the speed of microtubule shortening (*Efremov et al., 2007*; *Powers et al., 2009*).

Models in the second category, on the other hand, predict that a fraction of the force generated by a depolymerizing microtubule is harnessed to move a non-diffusive or poorly diffusive, tightly bound coupler to the next position on the microtubule lattice. This type of coupling is expected to result in reduced microtubule disassembly rates. For instance, this phenomenon has been observed in presence of oligomers of Dam1 (DASH) complexes, which bind tightly to the microtubule wall and may form microtubule-encircling rings (*Westermann et al., 2005*; *Miranda et al., 2005*; *Grishchuk et al., 2008*). In the model known as 'forced walk', tubulin power strokes generated at the base of the curls of disassemblying microtubules are necessary to move the ring into the next position of the lattice. The assembly of rings around the microtubule, however, is not a necessary condition for the exploitation of the energy of a depolymerizing microtubule, and other models, including models based on independent tight binders, have been proposed (*Asbury et al., 2011*; *Grishchuk, 2017*). For tightly bound couplers, therefore, a fraction of the microtubule-generated force is spent to move the coupler until the microtubule shortening is stalled by a combination of coupler's 'friction' and the opposing force (*Efremov et al., 2007*; *Grishchuk et al., 2008*).

Several observations indicate that the multivalent NDC80 modules, rather than undergoing biased diffusion as previously proposed (*Powers et al., 2009*), generate attachments that belong to the second category. The multivalent NDC80 modules generated a coupler that bound the microtubule tightly with residence in the range of hours (*Figure 2*), and that decreased microtubule depolymerization rates (*Figure 3*). In contrast, and consistent with previous studies (*Alushin et al., 2010*; *Schmidt et al., 2012*), monomeric NDC80 and monovalent $T_3S_1[NDC80]_1$ did not follow or modulate microtubule shortening (*Figure 1B*, 3BC). The stall forces produced by multivalent NDC80 modules saturated at 5–6 pN and increased the chance of microtubule rescue (*Figure 4*). Comparing these values to the theoretical maximum of 13–70 pN that a microtubule can generate (*Molodtsov et al., 2005*; *Vichare et al., 2013*), we speculate that a significant amount of microtubule-generated energy is spent to break the bonds between the TS-NDC80 assemblies and the microtubule. These

observations suggest a direct interaction between multivalent NDC80 assemblies and the microtubule tip, and make us suggest that these assemblies exploit the force of depolymerizing microtubules (*Figure 6*). This conclusion is consistent with recent experiments in vivo showing that kinetochore-oligomerized NDC80 is a friction element that slows down the speed of kinetochore movements in mitosis (*Long et al., 2017*).

Protofilaments at growing microtubule ends are less bent than at shortening ends, and force-induced straightening of protofilament flares has been suggested as a mechanism to convert the energy of microtubule depolymerization into cargo transport (*McIntosh et al., 2008*). Externally applied opposing force also slows down microtubule shortening (*Franck et al., 2007*), presumably through suppressed bending of depolymerizing protofilaments (*Grishchuk et al., 2005*). Thus, the reduced depolymerization rate of microtubules carrying TS-NDC80 assemblies supports a direct binding of multivalent NDC80 to flaring protofilaments (*Figure 3D*). We therefore propose that the correlation between the magnitude of the TS-NDC80 mediated stalling forces and the probability to reverse microtubule depolymerization into growth is a direct consequence of force-induced protofilament straightening. Kinetochore-associated fibrils that modulate the shape of depolymerizing microtubule ends at attached kinetochores have previously been identified in vivo using electron tomography (*McIntosh et al., 2008*), but their molecular identity has remained unclear. While our studies suggest that the fibrils might consist of oligomers of NDC80 complexes, further analyses will be required for a detailed molecular dissection. Importantly, previous structural analyses of an engineered version of the NDC80 complex (NDC80[Bonsai], *Ciferri et al., 2008*) identified this complex on the microtubule lattice (*Alushin et al., 2010*). However, until now it has been impossible to define the position of NDC80 complexes on a depolymerizing microtubule end under force, and therefore the question remains open.

Stabilization of kinetochore-microtubule attachments in vivo is tension-dependent (*Cane et al., 2013*; *Nicklas and Ward, 1994*). Here we reconstituted this behavior in vitro using dynamic microtubules and optically trapped modules with a defined NDC80 stoichiometry. The multivalency of NDC80 at the outer kinetochore thus likely directly contributes to the tension-dependent stabilization of kinetochore-microtubule in vivo. Stabilization of kinetochore-microtubule interactions by tension has also been recapitulated in vitro with purified kinetochore particles from *Saccharomyce cerevisiae* (*Akiyoshi et al., 2010*). These purified kinetochore particles, whose homogeneity and stoichiometry of microtubule binders has not been precisely defined, lose their connection to the microtubule ends under an externally applied force of several pN (*Akiyoshi et al., 2010*; *Miller et al.,*

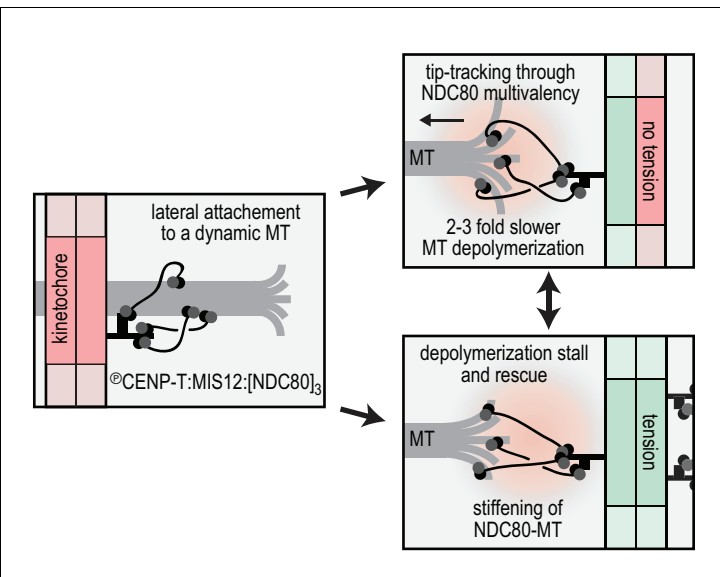

**Figure 6.** Reconstitution of a dynamic kinetochore-microtubule attachment. A graphical recapitulation of the kinetochore-microtubule interfaces reconstituted and characterised in this study and their occurrence in vivo.
DOI: https://doi.org/10.7554/eLife.36764.020

*2016*; *Sarangapani et al., 2014*). The force-coupling observed for the purified kinetochore particles appears to be crucially dependent on the Dam1 complex (*Akiyoshi et al., 2010*), an additional microtubule binder that contributes to the efficient force coupling at the microtubule plus-end in *Saccharomyce cerevisiae*. Dam1, as well as the functionally analogous human Ska complex, also confer plus-end tracking activity in vitro to monomeric NDC80 (*Janczyk et al., 2017*; *Lampert et al., 2010*; *Schmidt et al., 2012*; *Tien et al., 2010*; *Welburn et al., 2009*; *Helgeson et al., 2018*).

In conclusion, our work highlights that recapitulating stoichiometry and spatial arrangement of macromolecular assemblies is crucial for the bottom-up reconstitution and biophysical analysis of the kinetochore. The organization of NDC80 in multivalent modules is -at least partially- sufficient to recapitulate tension-dependent stabilization of cargo-microtubule coupling in vitro. How the constrained stoichiometry and spatial arrangement of NDC80, post-translational modifications, and additional cargo-couplers collectively modulate microtubule dynamics remains to be addressed in vivo and in vitro.

# Materials and methods

## Key resources table

| Reagent type (species) or resource | Designation | Source or reference | Identifiers | Additional information |
|---|---|---|---|---|
| Recombinant DNA reagent | pLIB | Peters laboratory, *Weissmann et al., 2016* | | addgene plasmid 80610 |
| Recombinant DNA reagent | pBIG1A | Peters laboratory, *Weissmann et al., 2016* | | addgene plasmid 80611 |
| Recombinant DNA reagent | pLIB NDC80 | Musacchio laboratory, *Huis In 't Veld et al., 2016* | | |
| Recombinant DNA reagent | pLIB NUF2 | Musacchio laboratory, *Huis In 't Veld et al., 2016* | | |
| Recombinant DNA reagent | pLIB SPC25-HIS | Musacchio laboratory, *Huis In 't Veld et al., 2016* | | |
| Recombinant DNA reagent | pLIB SPC24 | Musacchio laboratory, *Huis In 't Veld et al., 2016* | | |
| Recombinant DNA reagent | pBIG1A with NDC80C (SPC25-HIS) | Musacchio laboratory, *Huis In 't Veld et al., 2016* | | combined by biGBac cloning (*Weissmann et al., 2016*) |
| Recombinant DNA reagent | pLIB SPC25-SORT-HIS | Musacchio laboratory, this study | | |
| Recombinant DNA reagent | pLIB SPC24-SPY | Musacchio laboratory, this study | | |
| Recombinant DNA reagent | pBIG1A with NDC80C (SPC25-SORT-HIS) | Musacchio laboratory, this study | | combined by biGBac cloning (*Weissmann et al., 2016*) |
| Recombinant DNA reagent | pBIG1A with NDC80C (SPC25-SORT-HIS SPC24-SPY) | Musacchio laboratory, this study | | combined by biGBac cloning (*Weissmann et al., 2016*) |
| Recombinant DNA reagent | pGEX-6P GST-CENP-T 2–373 SORT | Musacchio laboratory, *Huis In 't Veld et al., 2016* | | |
| Recombinant DNA reagent | pBIG1A CDK1-GST: Cyclin-B1-HIS | Musacchio laboratory, *Huis In 't Veld et al., 2016* | | |
| Recombinant DNA reagent | pET21a Core Traptavidin | Howarth laboratory, *Chivers et al. (2010)* | | addgene plasmid 26054 |
| Recombinant DNA reagent | pET21a Dead Strepatavidin SpyCatcher | Howarth laboratory, *Fairhead et al. (2014)* | | addgene plasmid 59547 |
| Peptide, recombinant protein | MIS12C (DSN1-d100-109) | Musacchio laboratory, *Petrovic et al. (2016)* | | |
| Peptide, recombinant protein | Sortase 5M and Sortase 7M | Ploegh laboratory, see *Hirakawa et al. (2015)* | | addgene plasmids 51140 and 51141 |
| Peptide, recombinant protein | GGGGC-Alexa488 | ThermoFisher | | peptide for C-terminal sortase labeling |

*Continued on next page*

*Continued*

| Reagent type (species) or resource | Designation | Source or reference | Identifiers | Additional information |
|---|---|---|---|---|
| Peptide, recombinant protein | GGGGK-TMR | GenScript | | peptide for C-terminal sortase labeling |
| Peptide, recombinant protein | GGGGK-FAM | GenScript | | peptide for C-terminal sortase labeling |
| Software, algorithm | Kymo.m | Dogterom laboratory, this study | | Matlab script to trace fluorescent particles in kymographs |

## Protein expression and purification

Standard Gibson assembly or restriction-ligation dependent cloning techniques were used to generate pLIB vectors with NDC80, SPC25[SORT-HIS], SPC24, and SPC24[SPY]. Expression cassettes were combined on pBIG1 vectors using Gibson assembly as described (*Weissmann et al., 2016*). Baculoviruses were generated in Sf9 insect cells and used for protein expression in Tnao38 insect cells. Between 60 and 72 hr post-infection, cells were washed in PBS (10 mM $Na_2HPO_4$, 1.8 mM $KH_2PO_4$, 2.7 mM KCl, 137 mM NaCl, pH 7.4) and stored at $-80°C$. All subsequent steps were performed on ice or at 4°C. Thawed cells were resuspended in buffer A (50 mM Hepes, pH 8.0, 200 mM NaCl, 5% v/v glycerol, 1 mM TCEP) supplemented with 20 mM imidazole, 0.5 mM PMSF, and protease-inhibitor mix HP Plus (Serva, Germany), lysed by sonication and cleared by centrifugation at 108,000 g for 30 min. The cleared lysate was filtered (0.8 μM) and applied to a 5 ml HisTrap FF (GE Healthcare, Chicago, IL) equilibrated in buffer A with 20 mM imidazole. The column was washed with approximately 50 column volumes of buffer A with 20 mM imidazole and bound proteins were eluted in buffer A with 300 mM imidazole. Relevant fractions were pooled, diluted 5-fold with buffer A with 25 mM NaCl and applied to a 6 ml ResourceQ column (GE Healthcare) equilibrated in the same buffer. Bound proteins were eluted with a linear gradient from 25 mM to 400 mM NaCl in 30 column volumes. Relevant fractions were concentrated in 30 kDa molecular mass cut-off Amicon concentrators (Millipore, Burlington, MA) in the presence of additional 200 mM NaCl and applied to a Superose 6 10/300 column (GE Healthcare) equilibrated in 50 mM Hepes, pH 8.0, 250 mM NaCl, 5% v/v glycerol, 1 mM TCEP. Size-exclusion chromatography was performed under isocratic conditions at recommended flow rates and the relevant fraction were pooled, concentrated, flash-frozen in liquid nitrogen, and stored at $-80°C$.

GST-CENP-T[2-373] with a C-terminal -LPETGG extension was expressed in *E. coli* BL21(DE3)-Codon-plus-RIL cells and purified as described by PreScission cleavage from a GSTrap resin followed by Heparin affinity and size-exclusion chromatography (*Huis In 't Veld et al., 2016*). Purified CENP-T was C-terminally labeled using a sortase 5M mutant (*Hirakawa et al., 2015*) and GGGGC-Alexa488 (ThermoFisher, Waltham, MA) peptide and separated from sortase and the unreacted peptide by size-exclusion chromatography using a Superdex 200 10/300 column (GE Healthcare). CDK1:Cyclin-B1 was prepared and used as described (*Huis In 't Veld et al., 2016*) to phosphorylate the Alexa488-labeled CENP-T. Phosphorylated CENP-T was incubated with NDC80 and a truncated version of MIS12 that lacks the Dsn1[100-109] region (see *Petrovic et al., 2016*) and purified by size-exclusion chromatography using a superose 6 increase 5/150 column (GE Healthcare) equilibrated in 20 mM TRIS pH 8.0, and 200 mM NaCl, 2% v/v glycerol, and 2 mM TCEP. Fractions of 80 μL were collected, analyzed by SDS-PAGE followed by in-gel fluorescence and coomassie staining, and selected for further use as illustrated in *Figure 1—figure supplement 1*.

Core Traptavidin (T; addgene plasmid #26054) and Dead Streptavidin-SpyCatcher (S; addgene plasmid # 59547) were gifts from Mark Howarth (*Chivers et al., 2010*; *Fairhead et al., 2014*) and expressed in E. coli BL21 [DE3] RIPL (Stratagene, San Diego, CA). Expression was induced in cultures with an $OD_{600}$ of 0.9 by adding 0.5 mM IPTG for 4 hr at 37°C. Cells were washed in PBS and stored at $-80°C$. All subsequent steps were performed on ice or at 4°C. Cells were thawed in 1,5 volumes of lysis buffer (PBS with 10 mM EDTA, 1 mM PMSF, 1% Triton X-100, 0.1 mg/ml lysozyme), incubated for 60 min, lysed by sonication, and centrifuged at 10000 g for 30 min. The supernatant was discarded and pelleted material was resuspended in PBS with 10 mM EDTA and 1% Triton X-100, centrifuged as above, resuspended in PBS with 10 mM EDTA, and centrifuged again. Washed

inclusion bodies were resuspended in 6M Guanidine hydrochloride pH 1.0 and cleared at 21130 g for 10 min in 2 ml eppendorf tubes. Protein concentrations were determined using absorbance at 280 nm and the denatured T and S subunits were mixed in an approximate 1:2 molar ratio. Refolding into $T_xS_y$ tetramers was accomplished by dropwise dilution and an overnight incubation in a 100x volume of stirring PBS with 10 mM EDTA. This mixture was supplemented with 300 gr $(NH_4)_2SO_4$ per liter and crudely filtered using paper towels. $T_xS_y$ tetramers were precipitated by doubling the amount of added $(NH_4)_2SO_4$ and pelleted at 15000 g or 17000 g using a JA-14 or JA-10 rotor, respectively. The precipitate was resuspended in 50 mM Boric Acid, 300 mM NaCl, pH 11.0 and dialysed to 20 mM Tris pH 8.0 using a SnakeSkin membrane with a 7 kDa molecular mass cut-off (ThermoFisher). The $T_xS_y$ mixture was loaded onto a 25 ml Source 15Q anion-exchange resin and eluted in 20 mM Tris pH 8.0 with a linear gradient from 100 mM to 600 mM NaCl in eight column volumes at a flow rate of 1 ml/min. Relevant fractions were pooled, analyzed by SDS-PAGE followed by coomassie staining (boiled and not-boiled), and further purified if required by size-exclusion chromatography using a Superdex 200 10/300 column (GE Healthcare) equilibrated in 20 mM TRIS pH 8.0, and 200 mM NaCl, 2% v/v glycerol, 1 mM TCEP. Purified TS tetramers were concentrated using 10 kDa molecular mass cut-off Amicon concentrators (Millipore), flash-frozen in liquid nitrogen, and stored at −80°C.

## Assembly of TS-NDC80 modules

A mixture of NDC80 and $T_xS_y$ with a 3–4 fold molar excess of NDC80 per S subunit was incubated for 12–20 hr at 10°C in the presence of PMSF (1 mM) and protease inhibitor mix (Serva). The formation of $T_xS_y$-SPC24$^{SPY}_y$ was monitored using SDS-PAGE followed by coomassie staining (samples not boiled). We either used a sortase-labeled fluorescent NDC80 complex or included GGGGK-TMR peptide and a sortase 7M mutant (*Hirakawa et al., 2015*) in the overnight spy-coupling reaction. In the latter case, molar ratios of approximately 20 and 0.2 compared to NDC80 were used. Reaction mixtures were applied to a Superose 6 increase 10/300 or a Superose 6 increase 5/150 column (GE Healthcare) equilibrated in 20 mM TRIS pH 8.0, 200 mM NaCl, 2% v/v glycerol, 2 mM TCEP. Size-exclusion chromatography was performed at 4°C under isocratic conditions at recommended flow rates and the relevant fractions were pooled and concentrated using 30 kDa molecular mass cut-off Amicon concentrators (Millipore), flash-frozen in liquid nitrogen, and stored at −80°C.

## Low-angle metal shadowing and electron microscopy

TS-NDC80 assemblies were diluted 1:1 with spraying buffer (200 mM ammonium acetate and 60% glycerol) and air-sprayed as described (*Baschong and Aebi, 2006*; *Huis In 't Veld et al., 2016*) onto freshly cleaved mica pieces of approximately 2 × 3 mm (V1 quality, Plano GmbH, Germany). Specimens were mounted and dried in a MED020 high-vacuum metal coater (Bal-tec, Canonsburg, PA). A Platinum layer of approximately 1 nm and a 7 nm Carbon support layer were evaporated subsequently onto the rotating specimen at angles of 6–7° and 45° respectively. Pt/C replicas were released from the mica on water, captured by freshly glow-discharged 400-mesh Pd/Cu grids (Plano GmbH), and visualized using a LaB$_6$ equipped JEM-1400 transmission electron microscope (JEOL, Japan) operated at 120 kV. Images were recorded at a nominal magnification of 60,000x on a 4k × 4 k CCD camera F416 (TVIPS), resulting in 0.18 nm per pixel. Particles were manually selected using EMAN2 (*Tang et al., 2007*).

## Negative staining and electron microscopy

Taxol-stabilized microtubules were made by polymerizing 20 µM tubulin in the presence of 1 mM GTP at 37°C. Taxol was added to concentrations of 0.2, 2, and 20 µM after 10, 20, and 30 min respectively. Stabilized microtubules were sedimented over a warm MRB80 gradient with 40% glycerol, 1 mM DTT, and 20 µM taxol and were resuspended in MRB80 with 20 µM taxol. Microtubules were incubated for 15 min at room temperature with 10 nm biotin-nanogold (Cytodiagnostics, Burlington, ON) and $T_1S_3[NDC80]_3$. The 10 µL mixture with tubulin at 2 µM, $T_1S_3[NDC80]_3$ at 0.4 µM, and biotin-nanogold at 0.05 OD (520 nm maximum absorbance; 1000x diluted from the stock) was thereafter applied for 45 s to freshly glow-discharged 400 mesh copper grids (G2400C, Plano GmbH) with a continuous carbon film. Grids were washed three times with MRB80 buffer, once with freshly prepared 0.75% uranyl formate (SPI Supplies, West Chester, PA),

and then stained with the uranyl formate for 45 s. Excess staining solution was removed by blotting and the specimen was air-dried. Presented micrographs were recorded as described above.

## Tubulin and microtubules

Digoxigenin-labelled tubulin was produced by cycling of porcine brain extract in high-molarity PIPES (*Castoldi and Popov, 2003*) followed by labelling according to published protocols (*Hyman et al., 1991*). All other tubulins were purchased from Cytoskeleton Inc. GMPCPP-stabilized seeds were made by two rounds of polymerization to remove any residual GDP. 25 μM tubulin (40% dig-tubulin) supplemented with 1 mM GMPCPP (Jena Biosciences, Germany) were polymerized for 30 min at 37°C, spun down in Beckman Airfuge (5 min at 30 psi), resuspended in 75% of the initial volume of MRB80 (80 mM K-Pipes pH 6.9 with 4 mM $MgCl_2$ and 1 mM EGTA) and depolymerized on ice for 20 min. After that the solution was supplemented with 1 mM GMPCPP and polymerized again for 30 min at 37°C. Microtubule seeds were sedimented again, resuspended in 50 μL of MRB80 with 10% glycerol, aliquoted and snap-frozen in liquid nitrogen for storage at −80C for up to 2–3 months.

Taxol-stabilized microtubules were made by polymerizing 50 μM tubulin (8% dig-tubulin, 3–6% Hilyte-647 tubulin) in the presence of 1 mM GTP 30 min at 37°C, then 10–25 μM taxol was added for another 30–60 min. Polymerized tubulin was then sedimented in Beckman Airfuge (3 min at 14 psi) and resuspended in 50 μL MRB80 supplemented with 10 μM taxol.

## TIRF microscopy

Coverslips were cleaned in oxygen plasma and silanized as described (*Volkov et al., 2014*). Flow chambers were constructed by a glass slide, double-sided tape (3M) and silanized coverslip and perfused using a pipet. The chambers were first incubated with ~0.2 μM anti-DIG antibody (Roche, Switzerland) and passivated with 1% Pluronic F-127 (experiments with TS-NDC80) or 1% tween-20 (experiments with CENP-T-oligomerized NDC80). Then taxol-stabilized microtubules (300 μL diluted 1:30-1:600) were introduced followed by the reaction mix and the chambers were sealed with valap. The reaction mix contained MRB80 buffer with 1 mg/ml κ-casein, 10 μM taxol, 4 mM DTT, 0.2 mg/ml catalase, 0.4 mg/ml glucose oxidase and 20 mM glucose, supplemented with 10–35 pM of TS-NDC80. For experiments with dynamic microtubules the taxol microtubules were substituted with GMPCPP-stabilized microtubule seeds, and the reaction mix (short of taxol) was supplemented with 8 μM tubulin (4–6% labeled with HiLyte-647), 1 mM GTP and 0.1% methylcellulose. In all experiments the reaction mix was centrifuged in Beckman Airfuge for 5 min at 30 psi before adding to the chamber.

Imaging was performed at 30°C using Nikon Ti-E microscope (Nikon, Japan) with the perfect focus system (Nikon) equipped with a Plan Apo 100 × 1.45 NA TIRF oil-immersion objective (Nikon), iLas$^2$ ring TIRF module (Roper Scientific) and a Evolve 512 EMCCD camera (Roper Scientific, Germany). Images were acquired with MetaMorph 7.8 software (Molecular Devices, San Jose, CA). The final resolution was 0.16 μm/pixel. The objective was heated to 34°C by a custom-made collar coupled with a thermostat, resulting in the flow chamber being heated to 30°C.

## Image analysis

All images were analysed using Fiji (*Schindelin et al., 2012*). Kymographs were produced by a custom macro that creates an average projection perpendicular to a selected line through a reslice operation. Resulting kymographs were then analysed using a custom script in MatLab R2013b (MathWorks, Natwick, MA). Each horizontal line of the kymograph was fitted with a Gaussian function, with its peak being the central position of the fluorescent spot, and the area under the curve being the spot intensity (*Volkov, 2018*; copy archived at https://github.com/elifesciences-publications/kymo). Fluorescence intensity of the spot before the first bleach event was averaged to obtain the initial intensity of the spot. Height of the individual bleach event was determined by obtaining the bleaching traces of TS-NDC80 modules or CENP-T-Alexa488 molecules diffusing on taxol-stabilized microtubules, then smoothing these traces with the Chung-Kennedy filter as described (*Chung and Kennedy, 1991*; *Reuel et al., 2012*). Resulting smoothed traces were then used to build the histograms of all intensities that occurred during bleaching (*Figure 1E*) and further analysed as described (*Grishchuk et al., 2008*). Lifetime measurements were performed by calculating the time difference between the landing and detaching events in kymographs. For chambers containing trivalent and

tetravalent TS-[NDC80] modules or pre-assembled TN complexes, in which the oligomers were bound to the microtubules immediately after addition to the chamber, the amount of time between addition of the oligomer and start of imaging (1–3 min) was neglected, and spots already present on the microtubules in the first frame were considered as just landed.

## Preparation of beads coated with TS-NDC80 modules

1 µm glass COOH-functionalized beads (Bangs Laboratories, Fishers, IN) were suspended by sonication as 1% w/v in MES buffer (25 mM MES pH 5 supplemented with 0.05% tween-20), washed by centrifugation at 16 kG for 1 min, and then activated with EDC and Sulfo-NHS, each at 10 mg/ml, for 30 min at 23°C with vortexing. After three washes the beads were allowed to bind a mixture of 2 mg/ml PLL-PEG (Poly-L-lysine (20 kDa) grafted with polyethyleneglycole (2 kDa), SuSoS AG, Switzerland) with 0–10% v/v of PLL-PEG-biotin for 30 min at 23C. The reaction was quenched by adding 200 mM glycine. The beads were washed three times, resuspended at 0.2% w/v and stored at 4C.

Before each experiment, 10 µL of PLL-PEG-coated beads were washed using washing buffer (MRB80 with 2 mM DTT and 0.4 mg/ml casein), resuspended in 10 µL and mixed with 10 µL of NDC80C oligomer in the same buffer. Incubation was performed for 1 hr on ice with frequent pipetting, then the beads were washed three times and resuspended in 30 µL of washing buffer. Flow chambers with GMPCPP-stabilized seeds were prepared as described above, the reaction mix contained MRB80 buffer with 10–12 µM tubulin, 1 mM GTP, 1 mg/ml κ-casein, 4 mM DTT, 0.2 mg/ml catalase, 0.4 mg/ml glucose oxidase and 20 mM glucose. This reaction mix was centrifuged in Beckman Airfuge for 5 min at 30 psi, and then 1 µL of beads suspension was added to 14 µL of the reaction mix, added to the chamber and the chamber was sealed with valap. The final concentration of the beads was 0.004% w/v (80 fM, without losses).

To measure bead brightness, the beads were washed twice in MRB80 buffer, resuspended in 10 µL and a 4 µL drop was placed on a plasma-cleaned coverslip. The coverslip was then put on top of the slide and sealed with valap. Beads were imaged using brightfield and laser epifluorescence to get the positions and fluorescence intensities of all beads, respectively. Fluorescence intensity of a single fluorophore for normalization of bead fluorescence was obtained as described (*Volkov et al., 2014*).

## Laser tweezers and experiments with the beads

DIC microscopy and laser tweezers experiments were performed using a custom-built instrument described elsewhere (*Baclayon et al., 2017*). Images were captured using Andor (Ireland) Luca R or QImaging (Canada) Retiga Electro CCD cameras and MicroManager 1.4 software. At the start of each experiment 50 frames of bead-free fields of view in the chamber were captured, averaged, and used later for on-the-fly background correction. The images were acquired at eight frames per second, subjected to background substraction and each 10 consecutive frames were averaged.

Calibration of quadrant photo-detector (QPD) response was performed by sweeping the trapping 1064 nm beam with the trapped bead across the tracking 633 nm beam with acousto-optic deflector (AOD) over the distance of ±400 nm in two orthogonal directions. The central ±200 nm region of the resulting voltage-displacement curve was then fitted with a linear fit to determine the conversion factor. Stiffness calibration was performed by fitting the power spectral density as described (*Tolić-Nørrelykke et al., 2004*) with correction for the proximity of the coverslip (*Nicholas et al., 2014*). The axial position of the free bead in a trap was adjusted to leave 100–200 nm between the surfaces of the bead and the coverslip, while having the coverslip-associated microtubules in the focal plane of the objective. QPD signal was sampled at 100 or 10 kHz without additional filtering. All experiments were performed at 0.2–0.4W of the 1064 nm laser resulting in a typical trap stiffness of 0.015–0.033 pN/nm.

Force signals were analysed in MatLab R2013b. Direction of the force development was checked for consistency with video recording for each signal. Signals with ambiguous direction of the force, or beads attached to more than one microtubule were discarded. X and Y coordinates from the QPD recordings were then rotated to correspond to the directions along and across the microtubule. A portion of the signal corresponding to microtubule pulling was downsampled to 1 kHz and smoothed with a Chung-Kennedy filter (*Chung and Kennedy, 1991*; *Reuel et al., 2012*). The stall

force was determined as the difference between the stall level and the free bead level in the smoothed signal along the microtubule multiplied by trap stiffness, corrected for the nonlinear increase of the force as a function of the distance from the trap center (*Simmons et al., 1996*).

To calculate the effective stiffness of the link between the bead and the microtubule, we have measured the variance <var > of the signal along the microtubule during stall (for microtubules pulling on the bead), or during a pause in the piezo-stage motion (for control experiments). Stiffness was then calculated as $k_B T/$<var > . In control experiments for *Figure 4*, the tension was generated by first selecting the microtubule that was oriented along the Y-axis of the piezo-stage (parallel to the long side of the flow chamber), attaching the bead to the microtubule and then manual stepping in 10- or 100 nm steps with the piezo stage in the direction of the seed.

## Computer simulation of lifetimes

Simulations were performed using Gillespie algorithm (*Gillespie, 1977*) in MatLab R2013b. Each simulation run started with an oligomer with N binding sites with only one binding site being attached. In the case of N = 1 the only event that can occur afterwards is detachment with a fixed rate $k_{off}$ determined as 1/(average lifetime of $T_3 S_1 x[NDC80C]_1$)=0.62 $s^{-1}$. If N > 1, detachment of each 'attached' binding site happens at a rate $k_{off}$, and attachment of each available site inside the oligomer happens at a rate $k_{on}$. Simulation stopped when all binding sites in an oligomer transitioned to the 'detached' state. The time elapsed until the next event was determined by calculating the propensity towards this event using pre-generated random numbers.

## Acknowledgements

We thank I Stender and C Körner for assistance with protein purification, M Baclayon and R Dries for assistance with the optical trap setup, N Taberner for assistance with image analysis and M Kok and L Reese for suggestions and discussion regarding the modeling. PJH was supported by an EMBO short-term fellowship (grant 7203). MD acknowledges funding from the European Research Council Synergy Grant MODELCELL (grant 609822) and AM acknowledges funding from the European Research Council Advanced Grant RECEPIANCE (grant 669686) and the Deutsche Forschungsgemeinschaft Collaborative Research Centre (CRC) 1093.

## Additional information

### Competing interests

Andrea Musacchio: Senior editor, *eLife*. The other authors declare that no competing interests exist.

### Funding

| Funder | Grant reference number | Author |
|---|---|---|
| H2020 European Research Council | 669686 | Andrea Musacchio |
| H2020 European Research Council | 609822 | Marileen Dogterom |
| European Molecular Biology Organization | 7203 | Pim J Huis in 't Veld |
| Deutsche Forschungsgemeinschaft | CRC1093 | Andrea Musacchio |
| Max-Planck-Gesellschaft | Open-access funding | Andrea Musacchio |

The funders had no role in study design, data collection and interpretation, or the decision to submit the work for publication.

### Author contributions

Vladimir A Volkov, Conceptualization, Data curation, Formal analysis, Investigation, Visualization, Methodology, Writing—original draft, Project administration, Writing—review and editing; Pim J

Huis in 't Veld, Conceptualization, Investigation, Visualization, Methodology, Writing—original draft, Project administration, Writing—review and editing; Marileen Dogterom, Supervision, Funding acquisition, Project administration, Writing—review and editing; Andrea Musacchio, Conceptualization, Supervision, Funding acquisition, Writing—original draft, Project administration, Writing—review and editing

### Author ORCIDs
Vladimir A Volkov (ID) https://orcid.org/0000-0002-5407-3366
Pim J Huis in 't Veld (ID) http://orcid.org/0000-0003-0234-6390
Andrea Musacchio (ID) http://orcid.org/0000-0003-2362-8784

### Decision letter and Author response
Decision letter https://doi.org/10.7554/eLife.36764.023
Author response https://doi.org/10.7554/eLife.36764.024

## Additional files

### Supplementary files
• Transparent reporting form
DOI: https://doi.org/10.7554/eLife.36764.021

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
