## [Decision Letter]

[Editors’ note: a previous version of this study was rejected after peer review, but the authors submitted for reconsideration. The first decision letter after peer review is shown below.]

Thank you for submitting your work entitled "NDC80 clustering modulates microtubule dynamics under force" for consideration by *eLife*. Your article has been reviewed by two peer reviewers, and the evaluation has been overseen by a Reviewing Editor and a Senior Editor. The following individuals involved in review of your submission have agreed to reveal their identity: Bungo Akiyoshi (Reviewer #1).

Our decision has been reached after consultation between the reviewers and the Reviewing Editor. Based on these discussions we regret to inform you that your work will not be considered for publication in *eLife* at this stage.

The reviewers and Reviewing Editor all agreed that the application of SpyTag/SpyCatcher to produce defined multivalencies of Ndc80 was technically strong and will be an important system for studying kinetochore function. However, the consensus was that the paper is currently somewhat confirmatory and needs additional extensions of this system to merit publication in *eLife*.

The reviewers identified avenues to explore that they considered would make the paper suitable and these are given in the reviews below. It was felt that a minimal step would be to compare your Ndc80 clusters with a natural cluster formed downstream of CENP-T (point 1 of reviewer 1). The reviewers would expect you to do this in the various assays to compare their properties. As an alternative it would also be appropriate to investigate other aspects of Ndc80 function that go beyond the extensive prior literature (as discussed by reviewer 2).

The reviewers and Reviewing Editor debated whether to ask for these experiments during revision of the manuscript, but decided it was better to reject the paper at this stage. They would welcome a new submission if you can extend your assay to address the next level of complexity.

Reviewer #1:

Although it has been well established that multimerization of Ndc80 complexes is essential for their tip-tracking ability (e.g. McIntosh et al., 2008, Powers et al., 2009), no study systematically analyzed the role of multivalency. Taking advantage of a SpyTag/SpyCatcher system, the authors elegantly reconstituted well-defined modules containing 1, 2, 3, or 4 copies of Ndc80 complexes and analyzed their microtubule-binding activity and microtubule dynamics in TIRF and laser tweezer assays. They showed that at least two copies of Ndc80 complexes were required for tip tracking on disassembling microtubules, and also revealed a relationship between stall force and rescue frequency. Although many of the results reported in this manuscript are rather confirmatory, the application of the SpyTag/SpyCatcher system to examine the role of multivalency in biological systems is highly novel and important. I therefore support its publication in *eLife* if my following concerns can be addressed.

Essential revisions:

1) Although the use of SpyTag/SpyCatcher is an elegant way of reconstituting defined numbers of Ndc80 complexes, it is inevitably artificial. Given that the authors recently reconstituted a more physiological module containing two Ndc80 complexes on one CENP-T molecule (Huis in 't Veld et al., 2016), can they test this module and compare the result with T2S2?

2) Previous work reported much higher stall force (8-16 pN by (Driver et al., 2017), or ~25 pN by (Volkov et al., 2013)), compared to this manuscript (1.5 – 6 pN). It would be helpful if the authors could give clarification or possible explanation for why such a lower force stalled microtubule depolymerization in their assay.

3) Discussion section: I got an impression that the authors were arguing as if their data are inconsistent with the biased-diffusion model. However, their data could be interpreted as a support for the biased-diffusion model because, as they confirmed, Ndc80 requires multivalent attachment for tracking disassembling microtubules (which is an essence of the biased-diffusion model). In any case, it would be important to mention that forced walk and biased-diffusion models are not mutually exclusive.

4) Discussion section: "direct binding of clustered NDC80 to flaring protofilaments": This statement seems inconsistent with the authors' previous study (Alushin, 2010) that proposed preferential binding of Ndc80 to microtubule lattice rather than flaring protofilaments. A clarification would be helpful.

Reviewer #2:

This paper investigates the behavior of the kinetochore Ndc80 complex for its interactions with dynamic microtubules. Using an elegant strategy, the authors create conditions in which they can control the multimerization of the Ndc80 complex to test its behavior in the presence of 1, 2, 3, or 4 molecules. They find that increased Ndc80 numbers locally concentrated in this way result in the ability to track with depolymerizing microtubules and the production of modest forces. This paper is technically strong and is carefully conducted. However, the present manuscript essentially reproduces work from the Asbury lab (Powers et al., 2009) in which the authors artificially clustered Ndc80 molecules on beads. The prior work was not as carefully controlled for the precise numbers but came to the same overall conclusions for this behavior. Since then, this model of avidity has been the dominant way in which people have considered Ndc80 function. Thus, although the current paper is nicely executed, it does not provide conceptual advances that rise to the level that I consider for *eLife*. In their Introduction and Discussion section, the authors highlight multiple additional features of the kinetochore that are relevant to the binding, force, and movement behavior of the Ndc80 complex. This includes its upstream targeting factors and other associated proteins (such as the Dam or Ska complexes). There is also an extensive prior literature on regulatory modifications, and there has been a debate on the relative organization of Ndc80 molecules on microtubules. If the authors were able to build on their existing system and assay to evaluate these less well-defined features and properties of kinetochore-microtubule interactions, this would represent a more substantive advance.

[Editors’ note: what now follows is the decision letter after the authors submitted for further consideration.]

Thank you for submitting your article "Multivalency of NDC80 in the outer kinetochore is essential to track shortening microtubules and generate forces" for consideration by *eLife*. Your article has been reviewed by two peer reviewers, and the evaluation has been overseen by a Reviewing Editor and Jessica Tyler as the Senior Editor. The following individuals involved in review of your submission have agreed to reveal their identity: Bungo Akiyoshi (Reviewer #2).

The manuscript has been improved, but a few concerns remain. For your reference, we include the reviewers' comments below. Although there was some difference of opinion between the reviewers, both agreed that your paper makes a strong technical advance. Upon discussion they both came to the consensus that it should be eventually accepted. Before doing so, we would like you to address the two comments from reviewer #2. (There will be no need to review the work again.)

Reviewer #2:

In the revised manuscript, Volkov and colleagues addressed many of my concerns. Especially, they used more physiological complexes that consist of Ndc80 and CENP-T (with or without Mis12) and found a similar effect as the SpyCatcher-based experiment. Although the presented data are, as the authors admit, not of highest quality, it shows the importance of a better controlled system to analyze multivalency in single molecule experiments. I fully recommend the publication of this manuscript in *eLife*.

1) The experiment in Figure 1D is not controlled properly. Figure 1C should be repeated at 2 nM to exclude the possibility that the CENP-T:Mis12 complexes bind microtubules on their own at the concentration used in Figure 1D.

2) Figure 1—figure supplement 1D: What are the concentrations of TMN, TN, TM used in these TIRF experiments?

Reviewer #3:

This paper represents a careful technical execution and the creation of improved methods to test whether multimerizing the Ndc80 complex improves its microtubule tracking and force generating capabilities. The revised paper extends on the previous version where the authors used artificial oligomerization with defined Ndc80 numbers to oligomerize Ndc80 through its upstream receptor CENP-T. In this case, they similarly find improved properties when multiple Ndc80 complexes are closely associated. The authors have also revised aspects of the text in an attempt to better highlight their models and ideas.

Although I continue to find the level of experimentation impressive, I remain unconvinced that this paper represents a strong conceptual advance or new insights into the mechanisms of kinetochore-microtubule interactions. I do not believe that this paper is sufficient to resolve a debate between biased-diffusion or forced-walk models, or that it substantially alters the way that diverse researchers are considering the activities and properties of the Ndc80 complex. There have been a variety of both single molecule and force-based studies on Ndc80 behavior from the Asbury, Davis, Grishchuk, and Westermann labs (and many others). For example, the recent Davis lab paper (Helgeson et al.,) tests some related ideas on Ndc80 force and microtubule binding behavior, but also combining this with the Ska complex. In this current paper, the authors are unable to test the force properties of a single multimer, as they also need to attach these to beads.

I fully appreciate the incredible technical challenges required to conduct these type of experiments (as the authors highlight in their response letter). In this case, they may wish to publish a more methods-focused paper as a building step for a more extended work using this strategy to test various models and additional interacting factors as would be required to generate a more robust conceptual advance.

---

## [Author Response]

[Editors’ note: the author responses to the first round of peer review follow.]

Reviewer #1:Although it has been well established that multimerization of Ndc80 complexes is essential for their tip-tracking ability (e.g. McIntosh et al., 2008, Powers et al., 2009), no study systematically analyzed the role of multivalency. Taking advantage of a SpyTag/SpyCatcher system, the authors elegantly reconstituted well-defined modules containing 1, 2, 3, or 4 copies of Ndc80 complexes and analyzed their microtubule-binding activity and microtubule dynamics in TIRF and laser tweezer assays. They showed that at least two copies of Ndc80 complexes were required for tip tracking on disassembling microtubules, and also revealed a relationship between stall force and rescue frequency. Although many of the results reported in this manuscript are rather confirmatory, the application of the SpyTag/SpyCatcher system to examine the role of multivalency in biological systems is highly novel and important. I therefore support its publication in eLife if my following concerns can be addressed.Essential revisions:1) Although the use of SpyTag/SpyCatcher is an elegant way of reconstituting defined numbers of Ndc80 complexes, it is inevitably artificial. Given that the authors recently reconstituted a more physiological module containing two Ndc80 complexes on one CENP-T molecule (Huis in 't Veld et al., 2016), can they test this module and compare the result with T2S2?

We thank the reviewer for the constructive comments and the support of our work. We have now considerably extended the manuscript in the direction suggested by the reviewer. Specifically, we have added an entire figure (new Figure 1) and several additional supplemental panels in which we examine the interaction of NDC80 with microtubules in presence of phosphorylated and fluorescent CENP-T, plus or minus the MIS12 complex, thus creating scaffolds binding three or two NDC80 complexes, respectively. Inspection by electron microscopy revealed striking similarity of CENP-T complexes and TS constructs, as shown in a new panel in Figure 2. Monomeric NDC80 complexes behaved like (the monovalent) T3S1-[NDC80]_1_ constructs and did not follow the ends of shortening microtubules. The di- and trivalent NDC80 complexes on CENP-T and CENP-T:MIS12 were instead able to tip-track and reduced the depolymerization rates of shortening microtubules (Figure 1, Figure 1—figure supplement 1, Figure 3—figure supplement 1).

Thus, the new results prove that the TS-NDC80 modules recapitulate features of the “physiological” complexes of NDC80 with CENP-T and MIS12. This analysis also brought to light a very good reason for why TS modules may deliver more trustworthy results at this stage. At the picomolar concentrations required for single-molecule studies, the control of stoichiometry with the TS-based system was superior to that achievable with the CENP-T scaffold (see Figure 1F). This is likely a reflection of our inability to reproduce all stabilizing interactions on the CENP-T scaffold. On the other hand, the covalent bond forming between Spy-tagged NDC80 and Spy-catcher tagged streptavidin in the extremely stable tetrameric TS-modules is ideally suited to obtain well-defined stoichiometries at these low concentrations.

2) Previous work reported much higher stall force (8-16 pN by (Driver, et al., 2017), or ~25 pN by (Volkov et al., 2013)), compared to this manuscript (1.5 – 6 pN). It would be helpful if the authors could give clarification or possible explanation for why such a lower force stalled microtubule depolymerization in their assay.

We agree with the reviewer that this is an important question. At this stage, we can only speculate on possible reasons for this relatively low value of the stalling force. Most plausible is that we do not yet use fully reconstituted kinetochore particles. As a consequence (1) we miss the required multiplicity of binders (our 4 NDC80 complex assembly has half the number of NDC80 complexes that we expect to see on real kinetochores); (2) we miss the spectrum of potential binders, e.g. the SKA complex and other microtubule binders that act on real kinetochores. We feel that the last sentence in the Discussion section exposes this limit of the study quite fairly. While these considerations may appear to expose a limit of the work, we remain far from being able to carry out experiments with fully reconstituted kinetochores. A merit of the study is that it shows how using a well-controlled model can lead to fundamentally different conclusions from those that have been reached before.

3) Discussion section: I got an impression that the authors were arguing as if their data are inconsistent with the biased-diffusion model. However, their data could be interpreted as a support for the biased-diffusion model because, as they confirmed, Ndc80 requires multivalent attachment for tracking disassembling microtubules (which is an essence of the biased-diffusion model). In any case, it would be important to mention that forced walk and biased-diffusion models are not mutually exclusive.

We have now considerably streamlined the Discussion section to clarify how our observations fit within current models of kinetochore-microtubule attachment. In summary, current models for kinetochore-microtubule attachment fall under two main categories: biased diffusion and conformational wave models. While these models are not incompatible, as pointed out by the reviewer, all our observations converge on suggesting that Ndc80 multivalent particles harness the power stroke of depolymerizing microtubules: (1) they bind microtubule tightly; (2) they diffuse insignificantly on the lattice; (3) they stiffen the linkage with the microtubule under load; (4) they reduce microtubule depolymerizing rate; and (5) they promote microtubule rescue under load. We now clarify these points in the Discussion section and take a more clear standing about the significance of our data for this crucial on-going debate.

4) Discussion section: "direct binding of clustered NDC80 to flaring protofilaments": This statement seems inconsistent with the authors' previous study (Alushin, 2010) that proposed preferential binding of Ndc80 to microtubule lattice rather than flaring protofilaments. A clarification would be helpful.

We agree with the reviewer that previous structural work showed binding of NDC80 Bonsai complex along the microtubule lattice. We recognize two fundamental questions regarding this issue: (1) Do our multivalent assemblies recognize different features than the previously used monomeric Ndc80 assemblies (e.g. Bonsai); and (2) What position does NDC80 occupy when force opposes microtubule shortening? This is a condition that EM experiments have not yet been able to capture.

In principle, the power stroke of the microtubule may cause the detachment of individual NDC80 complexes from the microtubule flare, and their reattachment downstream along the lattice, as predicted by the “independent binder” model (see Grishchuck, 2017). We agree with the reviewer that our observations do not demonstrate NDC80 binding to flaring microtubules but are certainly consistent with it. We have now tried to clarify this point in the Discussion section, including the potential contradiction with the structural work.

Reviewer #2:This paper investigates the behavior of the kinetochore Ndc80 complex for its interactions with dynamic microtubules. Using an elegant strategy, the authors create conditions in which they can control the multimerization of the Ndc80 complex to test its behavior in the presence of 1, 2, 3, or 4 molecules. They find that increased Ndc80 numbers locally concentrated in this way result in the ability to track with depolymerizing microtubules and the production of modest forces. This paper is technically strong and is carefully conducted. However, the present manuscript essentially reproduces work from the Asbury lab (Powers et al., 2009) in which the authors artificially clustered Ndc80 molecules on beads. The prior work was not as carefully controlled for the precise numbers but came to the same overall conclusions for this behavior.

We thank the reviewer for these considerations. At the same time, we would like to express our respectful disagreement and bring arguments to object to certain parts of his/her criticism. In particular, we object that our work essentially reproduces the Powers et al., 2009 paper, and for two main reasons. First, Powers et al., 2009 did not describe experiments measuring the force-coupling properties of either monomeric or oligomerized NDC80. To our knowledge, ours is the first report that NDC80 alone, without additional factors like Dam1 or Ska, can capture force from depolymerizing microtubules. Second, and even more importantly, Powers et al., 2009 concluded that NDC80 moves by biased-diffusion without interacting with microtubule tips and without an effect on microtubule dynamics. We demonstrate that this is not the case: when physiologically relevant or artificial oligomers of NDC80 are used, a single NDC80 dimer or trimer is sufficient to slow down microtubule shortening. We further show that there is a unique connection between NDC80 and a dynamic microtubule end. The latter occurs specifically during stalled microtubule shortening and specifically when NDC80 oligomerized not through a bead, but through a protein-protein interaction scaffold. These and other observations we report are hardly compatible with biased-diffusion. Rather, our data indicate that kinetochore-mediated multivalency enables NDC80 to interact directly with flared protofilaments at the microtubule end.

We admit that our previous Abstract, Introduction, and Discussion section might have fallen short of communicating the significance of our conclusions in relation to previous work as effectively as they should have. We have therefore revisited these sections, where we now indicate explicitly that our data are incompatible with biased diffusion and why.

Since then, this model of avidity has been the dominant way in which people have considered Ndc80 function. Thus, although the current paper is nicely executed, it does not provide conceptual advances that rise to the level that I consider for eLife. In their Introduction and Discussion section, the authors highlight multiple additional features of the kinetochore that are relevant to the binding, force, and movement behavior of the Ndc80 complex. This includes its upstream targeting factors and other associated proteins (such as the Dam or Ska complexes). There is also an extensive prior literature on regulatory modifications, and there has been a debate on the relative organization of Ndc80 molecules on microtubules. If the authors were able to build on their existing system and assay to evaluate these less well-defined features and properties of kinetochore-microtubule interactions, this would represent a more substantive advance.

The reviewer is correct in pointing out that potential avidity of NDC80 is perceived as playing a role in kinetochore-microtubule attachment, but the extent of this role is highly debated. For instance, Zaytsev and colleagues attributed a modest role to NDC80 multivalency in their recent model of kinetochore-microtubule attachment. As a matter of fact, the effects of NDC80 multivalency have never been measured, and we regard this as a major advancement of our work.

We agree nevertheless with the reviewer that the work is incomplete, in the sense that it does not yet capture the full complexity of a kinetochore-microtubule interface, as already indicated in our response to reviewer #1. This is clearly stated in the last sentence of the Discussion section. It may sound defensive, but we also note that those who want to perform this type of experiments with rigour equivalent to that shown in this study will have to arm themselves with a lot of patience, because the technical challenges associated with generating the required quality of reagents and to emulate and validate all significant interactions is quite overwhelming.

For this reason, while we consider it premature at this stage, based on our assessment of reagent quality, to include other potential players like the SKA complex, the SKAP:Astrin complex, Kif18, CENP-E, or CENP-F, we have significantly extended the study in a very interesting direction indicated by reviewer #1. In particular, we have now included phosphorylated CENP-T with or without MIS12 complex as a physiologically relevant oligomerization platform for NDC80 complex. As noted above in our response to reviewer #1, phosphorylated CENP-T was sufficient to produce a tip-tracking particle with properties that were indistinguishable from streptavidin-oligomerized NDC80 but lacking the precise control over NDC80 oligomeric status (see newly added Figure 1, Figure 1—figure supplement 1 and Figure 2—figure supplement 1). Thus, we conclude that NDC80 binding to CENP-T, at least in our conditions, serves the only function: to oligomerize several NDC80 molecules into a particle with emerging microtubule-coupling properties that are not inherent to monomeric NDC80.

[Editors’ note: the author responses to the re-review follow.]

[…] Upon discussion they both came to the consensus that it should be eventually accepted. Before doing so, we would like you to address the two minor comments from reviewer #2. (There will be no need to review the work again.)Reviewer #2:1) The experiment in Figure 1D is not controlled properly. Figure 1C should be repeated at 2 nM to exclude the possibility that the CENP-T:Mis12 complexes bind microtubules on their own at the concentration used in Figure 1D.

We thank the reviewer for spotting this discrepancy. To address this concern, we repeated the single-molecule TIRF experiments with CENP-T:Mis12 (TM) and dynamic microtubules at higher concentrations of TM and did not observe TM binding to microtubules or following their ends. Figure 1C is now replaced to show a kymograph obtained at 2 nM TM, and a new panel in Figure 1—figure supplement 2 was added to demonstrate that TM does not interact with microtubules in the range of concentrations from 0.2 to 7 nM.

2) Figure 1—figure supplement 1D: What are the concentrations of TMN, TN, TM used in these TIRF experiments?

We are grateful to the reviewer for noticing this omission. Both TN and TMN are shown at 0.4 nM, this information is now added to the figure (new Figure 1—figure supplement 2A).